# Controlled Training Data Generation with Diffusion Models

**Teresa Yeo**[*] **Andrei Atanov**[*] **Harold Benoit**[†] **Aleksandr Alekseev**[†]
**Ruchira Ray** **Pooya Esmaeil Akhoondi** **Amir Zamir**
*Swiss Federal Institute of Technology Lausanne (EPFL)*

**Reviewed on OpenReview:** *https://openreview.net/forum?id=sSOxuUjE2o*

**Project page:** *adversarial-prompts.epfl.ch*

## Abstract

We present a method to control a text-to-image generative model to produce training data useful for supervised learning. Unlike previous works that employ an open-loop approach via pre-defined prompts to generate new data using either a language model or human expertise, we develop an automated **closed-loop** system that involves **two feedback mechanisms**. The first mechanism uses feedback from a given supervised model to find **adversarial prompts** that result in generated images that maximize the model's loss and, consequently, expose its vulnerabilities. While these adversarial prompts generate training examples curated for improving the given model, they are not curated for a specific target distribution of interest, which can be inefficient. Therefore, we introduce the second feedback mechanism that can optionally **guide** the generation process towards a desirable target distribution. We call the method combining these two mechanisms Guided Adversarial Prompts. The proposed closed-loop system allows us to control the training data generation for a given model and target image distribution (see Fig. 1 *(Right)*). We evaluate on different tasks, datasets, and architectures, with different types of distribution shifts (corruptions, spurious correlations, unseen domains) and illustrate the advantages of the proposed feedback mechanisms compared to open-loop approaches [1] .

## 1 Introduction

The quality of data plays a crucial role in training generalizable deep learning models (Taori et al., 2020; Miller et al., 2021; Gadre et al., 2023). For a model to generalize well, its training data should be representative of the test distribution where it will be deployed. However, real-world test conditions change over time, while training datasets are typically collected once and remain static due to high collection costs. This is also in contrast to evidence showing that being able to control the input data is a key contributor to how children are able to learn with only a few examples (Braddick & Atkinson, 2013; Lewis & Maurer, 2005). We, therefore, focus on generating training datasets that can adapt to novel test distributions and are more sample-efficient.

Diffusion generative models (Rombach et al., 2022; Ho et al., 2020; Sohl-Dickstein et al., 2015; Nichol et al., 2022; Saharia et al., 2022) are trained on large-scale collections of images (Schuhmann et al., 2022) and exhibit remarkable generalization abilities by being able to produce realistic images not seen during training. Additionally, unlike static datasets that they are trained on, these generative models allow us to *adapt* the generation process to produce images that follow a certain conditioning. For example, they can be conditioned on textual prompts (Rombach et al., 2022) or geometric information such as depth maps (Zhang & Agrawala, 2023).

Recent works explore the use of diffusion models to generate training data for supervised learning with promising results (Sariyildiz et al., 2023; Dunlap et al., 2023; He et al., 2022). They guide the generation process using text prompts to accomplish two goals: 1) produce aligned image-label pairs for supervised

---

[1]See the uploaded overview video.

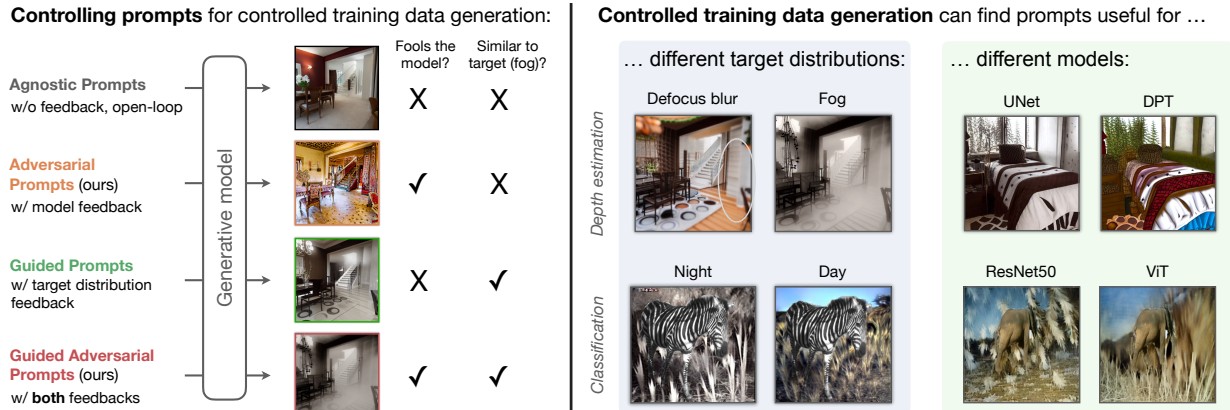

Figure 1: **We propose a framework for the automatic generation of training examples curated for a given model and target distribution.** ***Left:*** The method controls the prompts of a conditional text-to-image generative model via two feedback mechanisms. The 'model feedback' ensures that generated examples expose vulnerabilities by fooling a given model, resulting in Adversarial Prompts. The 'target distribution feedback' guides the generations towards a desirable target distribution (fog in this example.) Combining both feedback mechanisms results in Guided Adversarial Prompts, leading to generations that both fool the model and fit the desirable target distribution. ***Right:*** This approach allows us to find prompts that generate training examples useful for different target distributions, supervised models, and tasks (the oval highlights the effect of defocus blur in the corresponding example.) While the differences in generations for different models might appear subtle or unintuitive, they, indeed, result in more useful training data (see Sec. 4.3).

training and 2) adapt the generated images to a certain target distribution. However, these methods find conditioning text prompts in an open-loop way, i.e., uninformed of the specific training scenario in hand, by either using a language model (Dunlap et al., 2023) or heuristics (Sariyildiz et al., 2023). In other words, they *lack an automatic feedback mechanism* that can refine the found text prompts to produce more curated and useful training data.

In this work, we propose two feedback mechanisms to find prompts for generating useful training data. The first mechanism finds prompts that result in generations that maximize the loss of a particular supervised model, thus, *reflecting its failure modes*. We call them Adversarial Prompts (AP). This mechanism ensures that we find not only novel prompts, which may still produce images that the model already performs well on, but adversarial prompts that produce images with high loss, and, thus, useful for improving the model (Deng et al., 2021) (see exemplar generations for AP in Figs. 1, 3 and 5)

One given model can perform poorly on *multiple* distribution shifts, and hoping to cover all of the shifts via Adversarial Prompts is an inefficient strategy (e.g., see the difference between the AP generation and the target distribution, fog, in Fig. 1 *(Left)*). Therefore, we introduce an additional *target distribution-informed* feedback mechanism that can find prompts that generate images fitting a desirable target distribution we want to adapt to. To implement it, we assume access to a textual description of the target distribution and/or *a few unlabeled* images from it. We then optimize a similarity metric between CLIP (Radford et al., 2021) embeddings of the generated examples and the target description. We call prompts that combine both mechanisms Guided Adversarial Prompts (GAP). Compare the Adversarial Prompts and Guided Adversarial Prompts in Figs. 1 and 3 to 5 to see the effect of the guidance feedback in steering the generations towards a specific target distribution.

We perform evaluations on different tasks (image classification, depth estimation), datasets with distribution shifts (Waterbirds (Sagawa et al., 2019), iWildCam (Beery et al., 2021; Koh et al., 2021), Common Corruptions (Hendrycks & Dietterich, 2019), 3D Common Corruptions (Kar et al., 2022)), and architectures (convolutional and transformer) and show supportive results.

## 2   Related Work

**Open-loop data generation** methods use pre-defined controls to guide the generative process and produce novel training examples. One line of work uses GANs (Jahanian et al., 2021; Chai et al., 2021; Ravuri & Vinyals, 2019) and pre-define a perturbation in their latent space to generate novel examples. More recent works adopt text-to-image diffusion models and use pre-defined prompt templates (Sariyildiz et al., 2023; Yuan et al., 2022; He et al., 2022) or use a language model to generative variations of a given prompt (Yuan et al., 2022). These methods require *anticipating the kind of data that will be seen at test-time* when defining the prompts. On the other hand, our CLIP guidance mechanism allows us to generate images similar to the target distribution. Dunlap et al. (2023) also approach this problem by using a captioning and language model to summarize a target distribution shift into a text prompt. However, this summarization process is not informed of the generations, and, thus, does not guarantee that the text prompt will guide the generation process to images related to the target distribution. Finally, these methods are not model-informed and do not necessarily generate images *useful* for training a given model.

**Closed-loop data generation** methods guide the generation process via an automatic feedback mechanism. They control the latent space of GANs (Besnier et al., 2020) or VAEs (Wong & Kolter, 2020) models, NeRF (Dong et al., 2022), or the space of hand-crafted augmentations (Cubuk et al., 2018) to generate data that maximizes the loss of the network on the generated data. Similarly, Jain et al. (2022) uses an SVM to identify the failure modes of a given model and uses this information to generate training data with a diffusion model. Our method employs a similar adversarial formulation (in conjunction with target distribution guidance) but performs the optimization in the text prompt space of recently developed diffusion models.

**"Shallow" data augmentation** techniques apply simple hand-crafted transformations to training images to increase data diversity and improve the model's generalization. Examples of such transformations are color jitter, random crop, and flipping, etc. To produce more diverse augmentations, methods like RandAugment (Cubuk et al., 2020) and AugMix (Hendrycks et al., 2019) combine multiple of such simple transformations, and Mixup (Zhang et al., 2017) and CutMix (Yun et al., 2019) methods use transformations that can combine multiple images. AutoAugment (Cubuk et al., 2018) and adversarial training (Madry et al., 2017) build a closed system to tune the parameters of the applied augmentations but are inherently limited by the expressiveness of the simple transformations. In contrast, our method uses expressive diffusion models, which results in images that are more diverse and realistic than those produced by "shallow" augmentations.

**Controlling diffusion models.** Methods like ControlNet (Zhang & Agrawala, 2023) and T2I-Adapter (Mou et al., 2023) adapt a pre-trained diffusion model to allow for additional conditioning e.g., edge, segmentation, and depth maps. We employ these models for generation as it allows us to generate paired data for different tasks, given the labels from an existing dataset. Editing methods aim to modify a given image, either via the prompt (Hertz et al., 2022), masks (Couairon et al., 2022), instructions (Brooks et al., 2023) or inversion of the latent space (Mokady et al., 2023; Huberman-Spiegelglas et al., 2023). In contrast, personalization methods aim to adapt diffusion models to a given concept e.g., an object, individual, or style. Popular examples include textual inversion (Gal et al., 2022) and DreamBooth (Ruiz et al., 2023), which aim to find a token to represent a concept given several images of that concept. The former freezes the diffusion model, while the latter fine-tunes it. Extensions of these works learn to represent multiple concepts (Avrahami et al., 2023; Han et al., 2023). In our work, we adopt an approach similar to textual inversion to steer the diffusion model, but our method can also be used with other controlling mechanisms.

## 3   Method

We begin this section by formalizing our problem setting and describing how diffusion models can be used to generate training data (Sec. 3.1). We then introduce two feedback mechanisms to find prompts that are informed of the failure modes of a given model (Sec. 3.2) and relevant to a given target distribution (Sec. 3.3).

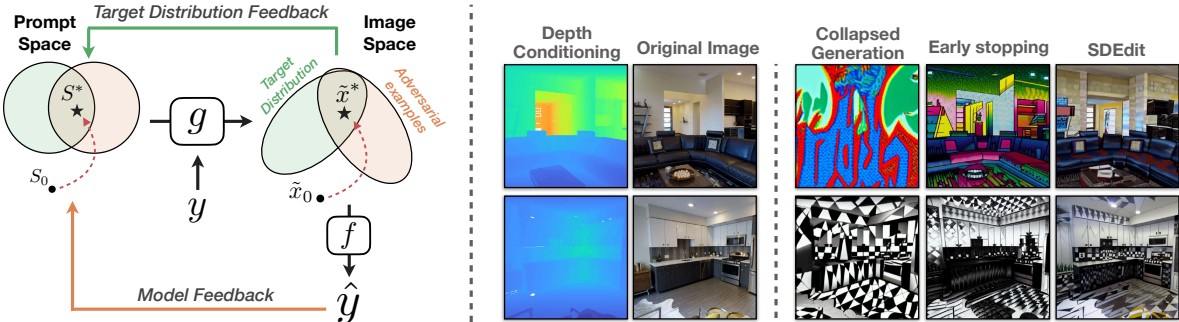

Figure 2: **Left: An overview of how we generate training data** for a given supervised model $f$ and target distribution. Suppose $g$ is generative model that generates images conditioned on a text prompt, $S$ and label, $y$. A supervised model trained to perform a task, e.g., image classification or depth estimation, is denoted by $f$. We aim to find prompts that would generate training data useful for $f$, and we do so via two feedback mechanisms. The first mechanism makes use of feedback from $f$ to get Adversarial Prompts. Since there can be many adversarial examples not relevant to a particular target distribution, we introduce a second feedback mechanism that guides the prompts towards a certain target distribution. Combining the two mechanisms results in Guided Adversarial Prompts. This results in generations that are both *relevant to the target distribution* and where *f does not perform well*. **Right: Ways to alleviate the misalignment of the generation with its conditioning.** The third column onwards shows some examples of generations from depth maps that *do not follow* the depth conditioning. See the first and second columns for the original image and its depth label. There are several ways to constrain the generation to alleviate this misalignment. **1.** Early stopping involves stopping the adversarial optimization when the loss reaches a certain threshold. The resulting generations from early stopping are shown in the fourth column. **2.** SDEdit (Meng et al., 2021b) involves conditioning the generation process on the original image. This mechanism is applied during generation with the adversarial prompts i.e., applying SDEdit to the prompts that generated the images in the third column results in the last column generations. Both SDEdit and early stopping are able to improve the alignment of the generations with depth conditioning.

### 3.1 Preliminaries

**Problem Formulation.** We consider the supervised learning problem, where a model $f : \mathcal{X} \rightarrow \mathcal{Y}$ learns a mapping from the image space $\mathcal{X}$, to a target space $\mathcal{Y}$, e.g., depth estimation or semantic classification. The model $f$ is trained using a training dataset $\mathcal{D}_{\text{train}}$ and tested on a new set $\mathcal{D}_{\text{test}}$ that exhibits a distribution shift w.r.t. the training data, e.g., corruptions Hendrycks & Dietterich (2019). Our goal is to generate additional *synthetic* training data $\mathcal{D}_{\text{syn}}$ to adapt the model and improve its performance under the distribution shift.

**Text-to-image Diffusion Models.** We use the Stable Diffusion (Rombach et al., 2022) text-to-image diffusion model as the basis for our generator $g$. Given a textual prompt $c$, Stable Diffusion is capable of synthesizing realistic images following the textual conditioning. However, in general, for a given task, e.g., depth estimation, a textual prompt alone may not be sufficient for controlling the generation well enough to produce aligned image-label examples.

**Generating aligned training examples.** We employ the following two approaches to condition the generative model $g$ on a label $y$ from the given training datasets and sample aligned training examples $(\tilde{x}, y)$. For the depth estimation task, we use the ControlNet (Zhang & Agrawala, 2023) model which extends the conditioning mechanisms of the Stable Diffusion to accept various spatial modalities, e.g., depth maps and segmentation masks. Specifically, we use ControlNet v1.0 with depth conditioning[2]. For semantic classification tasks, we utilize the foreground object masks and use an in-painting technique proposed by Lugmayr et al. (2022) that preserves the masked region throughout the denoising process, essentially keeping it intact. These mechanisms provide us with a generative model conditioned both on a text prompt $c$ and label $y$. We denote the resulting distribution modeled by this generative model as $g(y, c)$.

---

[2]https://github.com/lllyasviel/ControlNet

## 3.2 Model-Informed Generation with Adversarial Prompt Optimization

Our first feedback mechanism aims at generating training examples that reflect the failure modes of a given model $f$. An automatic way to do so is via **adversarial optimization**, which finds the "worst case" failure modes of $f$. More precisely, we find a prompt $c$ that generates images $\tilde{x} \sim g(y, c)$ that maximize the supervised loss $\mathcal{L}(f(\tilde{x}), y)$, e.g., $l_1$ loss for depth estimation. Since the usual prompt space is discrete (text tokens) and challenging to optimize over, we employ the approach introduced by Gal et al. (2022) and, instead, optimize over the corresponding continuous embedding space. For ease of notation, "prompt space" will implicitly refer to the continuous embedding space instead of the discrete token space. We construct a prompt $c_w$ out of $n$ new "placeholder" tokens, i.e., $c_w = (c_{w_1}, \ldots, c_{w_n})$, and find their corresponding embedding weights $\{w_i\}_{i=1}^n$ by solving the following optimization problem:

$$w_{\mathrm{AP}} = \arg \min_w \mathbb{E}_y \; \mathbb{E}_{\tilde{x} \sim g(y, c_w)} \mathcal{L}_{\mathrm{adv}}(f(\tilde{x}), y), \tag{1}$$

where $\mathcal{L}_{\mathrm{adv}} = -\mathcal{L}$ and $y$ is sampled from $\mathcal{D}_{\mathrm{train}}$. Note that the sample $\tilde{x}$ is differentiable w.r.t. the embeddings $w$ which allows us to use gradient-based optimization. We call the prompts that result from solving the above optimization problem Adversarial Prompts (AP).

**Avoiding $(\tilde{x}, y)$ alignment collapse.** While the adversarial objective in Eq. (1) aims to fool the model $f$, it may instead fool the label-conditioning mechanism of the generative model $g$, resulting in $c_{w_{\mathrm{adv}}}$ generating samples $\tilde{x} \sim g(y, c_{w_{\mathrm{adv}}})$ that are not faithful to $y$ (see Fig. 2 *(Right)*). To avoid this collapse, we use the following techniques.

First, we limit the expressivity of the generative model using the SDEdit method (Meng et al., 2021a). It conditions the generation process on the original image $x$ by starting the denoising process from its noised version instead of pure noise, constraining the expressive power of the generative model and producing samples closer to the original image $x$, and, therefore, more faithful to $y$.

Additionally, we implement constraints w.r.t. $\mathcal{L}_{\mathrm{adv}}$. For depth estimation, we employ an early stopping criterion and stop the adversarial optimization when the loss reaches a certain threshold. We also experimented with constraining the embedding weights $w$ to be close to existing vectors from the vocabulary in E.q. 1. However, we found that this significantly reduces the expressivity of the generations. Thus, we do not constrain $w$ during optimization. For image classification, we found that adversarial optimization of the standard cross-entropy loss can lead to the generation of images of another class (e.g., generating elephant, when $y =$ giraffe). We, therefore, maximize the entropy of the model's predictions, i.e., its uncertainty, instead, which we found to lead to the generation of aligned $(\tilde{x}, y)$ examples. Please see Appendix B.5.2 for more details and examples.

## 3.3 Target Distribution Informed Generation

The adversarial formulation above finds prompts that reflect the failure modes of $f$. Without any information about the target distribution, improving the model on the worst-performing distributions is one of the best strategies, which, indeed, improves performance in multiple cases (see Fig. 4 and Tab. 1). However, a given model typically has many failure modes and many possible distribution shifts that can occur at test time. Adapting to all of them using only the adversarial feedback mechanism might be inefficient if the goal is to adapt to a specific target distribution instead of improving average performance. Thus, we introduce the second feedback mechanism to inform the prompt optimization process of the target image distribution. It only requires access to simple text descriptions (e.g., 'fog' to adapt to foggy images) or a small number ($\sim 100$) of *unlabelled* images.

We implement the target-informed feedback mechanism using CLIP (Radford et al., 2021) guidance. Specifically, we assume access to either textual descriptions of the target image distribution $\{t_j\}$, a few unlabeled image samples $\{x_j\}$, or both. We construct the corresponding text and image guidance embeddings as $e_t = \mathrm{avg}(\{E_t(t_j)\})$ and $e_i = \mathrm{avg}(\{E_i(x_j)\})$, where $E_t$ and $E_i$ denote, respectively, the CLIP text and image encoders, and avg stand for averaging. We then use the following guidance loss:

$$\mathcal{L}_{\mathrm{G}}(\tilde{x}, c_w) = \lambda_t \mathcal{L}_t(E_t(c_w), e_t) + \lambda_i \mathcal{L}_i(E_i(\tilde{x}), e_i), \tag{2}$$

where we take $\mathcal{L}_t$ to be $l_2$ norm between two embeddings and $\mathcal{L}_i$ to be the negative cosine similarity, as we found it to perform the best. See the Appendix B.7 for the results of this ablation. Note that our formulation allows the use of only text or only images for guidance (setting corresponding $\lambda$s to 0) based on the available information. In Appendix B.6, we also show that one can construct a similarly effective guidance mechanism without relying on the CLIP model.

Finally, we combine both adversarial, Eq. (1), and guidance, Eq. (2), losses to form the final objective:

$$w_{\mathrm{GAP}} = \arg\min_w \mathbb{E}_y \; \mathbb{E}_{\tilde{x} \sim g(y, c_w)} \left[ \mathcal{L}_{\mathrm{adv}}(f(\tilde{x}), y) + \mathcal{L}_{\mathrm{G}}(\tilde{x}, c_w) \right]. \tag{3}$$

We call the prompts that result from solving Eq. (3), Guided Adversarial Prompts (GAP). See the Appendices B.4.2 and B.5.3 for further implementation details.

## 4 Experiments

We perform experiments in three settings: domain generalization via camera trap animal classification on the iWildCam (Beery et al., 2021) dataset, bird classification with spurious correlation with the Waterbirds (Sagawa et al., 2019) dataset, and depth estimation with the Taskonomy dataset (Zamir et al., 2018b; 2020). For depth estimation, we evaluate on distribution shifts from Common Corruptions (Hendrycks & Dietterich, 2019) (CC), 3D Common Corruptions (Kar et al., 2022) (3DCC) applied on the Taskonomy (Zamir et al., 2018b) test set and cross dataset shift from the Replica (Straub et al., 2019) dataset.

### 4.1 Semantic Image Classification

**Waterbirds** (Sagawa et al., 2019) is a dataset constructed by pasting an image of a waterbird or landbird from the CUB (Wah et al., 2011) dataset, which represents the label $y$, onto a background image of "land" or "water" from the Places (Zhou et al., 2014) dataset. We follow Dunlap et al. (2023) and use only images of waterbirds on water and landbirds on land as $\mathcal{D}_{\mathrm{train}}$, i.e., the background is completely correlated with the label in training data (see Fig. 3-right). The test set $\mathcal{D}_{\mathrm{test}}$ contains all four combinations of bird types and backgrounds (see Fig. 7).

**iWildCam** (Beery et al., 2021; Koh et al., 2021) is a domain generalization dataset made up of images captured from camera traps placed in various locations around the world. The goal is to generalize to photos taken from new camera deployments. We follow Dunlap et al. (2023) and create a 7-way classification task (background, cattle, elephant, impala, zebra, giraffe, dik-dik), use two locations that are not present in the training or validation as $\mathcal{D}_{\mathrm{test}}$, and fix the number of additional generated images for finetuning to 2224.

For both datasets, we use the ResNet50 (He et al., 2016a) model. We compare the following methods for generating additional synthetic data $\mathcal{D}_{\mathrm{syn}}$ (we provide more details in Appendix B):

*No Extra Data*: Only $\mathcal{D}_{\mathrm{train}}$ is used, without generating extra data.

*Augmentation baselines*: We use two data augmentation methods commonly used in recent literature: CutMix (Yun et al., 2019) and RandAugment (Cubuk et al., 2020). These baselines do not use generative models.

*Agnostic Prompts*: We use a prompt that is uninformed of both the model and the specific target distribution. Similar to ALIA (Dunlap et al., 2023), we use a prompt *"nature"* for Waterbirds and the prompt template "*a camera trap photo of {class name}*" for iWildCam.

*Guided Prompts*: We compare to ALIA (Dunlap et al., 2023), which uses captioning and language models to summarize a target distribution shift into text prompts. This results in seven prompts for Waterbirds and four prompts for iWildCam. See Appendix B.2 for more details, specific prompts, and a discussion on the differences between ALIA and our method.

*Adversarial Prompts*: We use the model previously trained on $\mathcal{D}_{\mathrm{train}}$ (i.e., no extra data) as the target model $f$ and find adversarial prompts following Eq. (1). We find four prompts per class for Waterbirds, eight in total, and four prompts in total applied to all classes for iWildCam.

*Guided Adversarial Prompts*: We use the same setting as in Adversarial Prompts and apply additional CLIP guidance to adapt to a target distribution shift, following Eq. (3). For Waterbirds, we apply text guidance

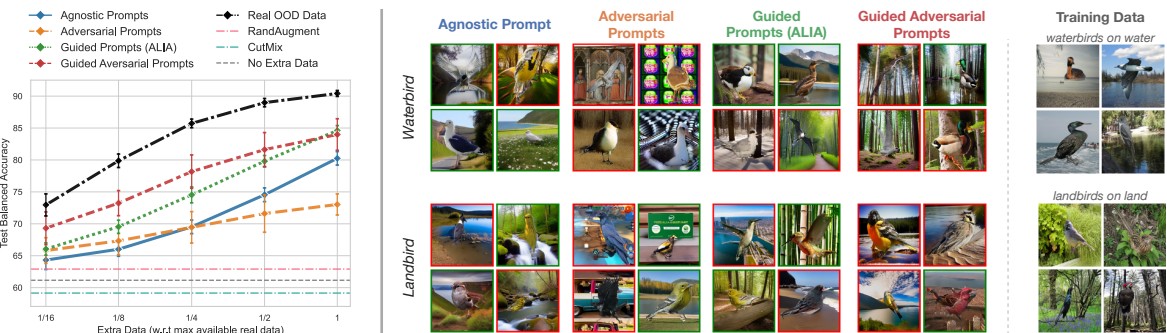

Figure 3: **Guided Adversarial Prompts generate counterfactual examples not present in the original training dataset and achieve better data efficiency than other prompts.** *Left:* Test performance on a balanced test set (waterbirds and landbirds appear on both land and water) of a model trained on the original spuriously correlated dataset, while varying the number of extra data points generated by different types of prompts (see Appendix C.2.3 for results on 2X data). We run each experiment with three seeds and report the mean and standard deviation. *Right:* Exemplary generations for each prompt type and original training examples (where background is correlated with the bird type.) Frames show whether the model trained without extra data made a correct prediction (green) or not (red) on the corresponding image. As expected, the adversarial mechanism produces examples that fool the model, and the guidance mechanism steers the generation toward the target distribution. Combining both mechanisms in GAP leads to the **generation of counterfactual examples** (waterbirds on land and landbirds on water), missing in the training dataset, and improved data efficiency compared to only target-informed GP.

using ALIA prompts as the textual description of the target distribution. For iWildCam, we use image guidance and partition the target test distribution into four groups based on two attributes that have a significant impact on the visual characteristics of the data: the test location (first or second) and time of the day (day or night). We sample 64 *unlabelled* images randomly from each group. We optimize for one guided adversarial prompt per group, resulting in four prompts in total.

**Training details.** For supervised training, we follow Dunlap et al. (2023) and use the ResNet50 model for $f$. In Sec. 4.3, we also explore how adversarial prompts change with the model by using ViT (Dosovitskiy et al., 2020). For prompt optimization, we use Adam (Kingma & Ba, 2017) with `1e-3` learning rate. For efficiency, we use five denoising steps during optimization and 15 for the final generations, both using the DDIM (Song et al., 2020) scheduler. As $\mathcal{L}_{\mathrm{adv}}$, we use prediction's entropy for iWildCam to avoid alignment collapse between $\tilde{x}$ and $y$ as discussed in Sec. 3.2, and negative cross-entropy for Waterbirds where we did not observe collapse. Please see Appendix B for more details on the experimental setup.

**Waterbirds results.** Fig. 3 *(Left)* shows the performance of each method, and Fig. 3 *(Right)* demonstrates the corresponding generated examples for each method. First, we find that while the performance of Adversarial Prompts is on par with Agnostic Prompts in a low-data regime, it performs worse with more generated data. Looking at the corresponding generated images, we find that while being adversarial, they appear different from images from the target distribution, which can explain the inferior performance in this particular case. Second, using Guided Prompts informed of the target distribution leads to a consistent improvement over Agnostic Prompts, and the corresponding images look more similar to the real images from the target distribution. Finally, combining both feedback mechanisms in Guided Adversarial Prompts results in more data-efficient generations, outperforming all other methods in the low-data regime. Looking at the corresponding generations, we find that GAP tends to generate only combinations of the bird type and background that are missing in the training dataset, i.e., waterbirds on land and landbirds on water, which can explain its higher sample efficiency.

**iWildCam results.** Fig. 4 shows that similar to Waterbirds, GAP, combining both feedbacks, achieves the best performance. Compared to GP with the target-only feedback, GAP generates images with a "camouflage" effect that fools the classification model. Compared to AP with the model-only feedback that generates images with snow background, GAP's images are more similar to the target domain that is summarized as "grassy field" (by GPT-4 in (Dunlap et al., 2023)). We also find that in the case of iWildCam, AP significantly

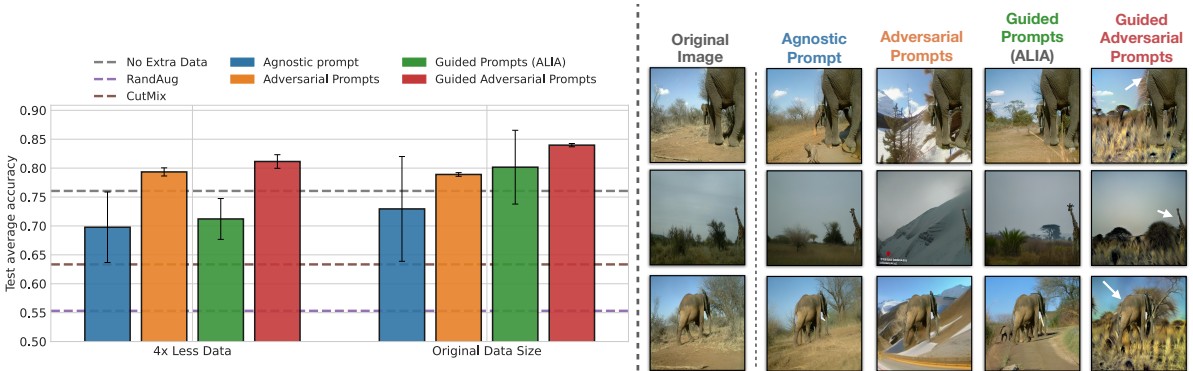

Figure 4: **Guided Adversarial Prompts generate hard training examples leading to the best performance.** *Left:* Performance on the images from unseen camera locations (out-of-distribution) with different numbers of generated synthetic images for iWildCam. We repeat each experiment three times and report the mean and standard deviation. GAP consistently outperforms other types of prompts. We also find that AP significantly outperforms GP in the low-data regime, suggesting that being only target-informed requires a more exhaustive sampling to find useful training examples. *Right:* Original training examples from iWildCam and generated examples for each type of prompt. For GAP, arrows point to the animals for clarity; from top to bottom, they are misclassified as cattle, dik-dik, cattle. GAP generates images that are both visually coherent with the target location (described as "grassy field") and introduce an adversarial "camouflage" effect.

outperforms GP in the low-data regime, suggesting that model-informed feedback generates more useful training examples. We also find that AP and GAP outperform standard (non-generative) augmentation baselines and domain generalization baselines (see Appendix C.2.2 for results).

## 4.2   Depth Estimation

We use the Taskonomy (Zamir et al., 2018a) and Omnidata Eftekhar et al. (2021a) datasets for training. The former consists of 4 million images of indoor scenes, while the latter is a mixture of datasets and contains both indoor and outdoor scenes. We evaluate our method on a range of domain shifts, from Common Corruptions (Hendrycks & Dietterich, 2019), 3D Common corruptions (Kar et al., 2022) to other datasets like Replica. We compare the following methods. They all involve fine-tuning the originally trained model $f$ on different synthetically generated datasets. See Fig. 5 for a comparison of the generations:

*Control (No extra data)*: We fine-tune $f$ on the original training data. This baseline is to ensure that the difference in performance is due to the generated data rather than, e.g., longer training or other hyperparameters used during fine-tuning.

*Agnostic Prompts*: This baseline generates data that is *agnostic* to the model and the target distribution. We generate images with the prompt "*room*" as the datasets consist of indoor images from mostly residential buildings.

*Agnostic Prompts (Random)*: As we use 30 tokens for our methods, this baseline controls for the number of tokens used. Here, we sample the same number of tokens as AP randomly from the vocabulary to construct a prompt. See Appendix C.1 for details.

*Adversarial Prompts*: We perform the optimization as described in Eq. (1), to get 30 adversarial prompts (tokens).

*Guided Prompts*: We generate data using only the CLIP guidance loss described in Eq. (2). We consider both image and text guidnace. For the former we use about 100 *unlabelled* images from the target distribution, for the latter we use the name of the shift e.g., "fog".

*Guided Adversarial Prompts*: We combine adversarial and guidance losses. We use the same settings as Adversarial Prompts and apply CLIP guidance loss used for Guided Prompts. This allows us to generate data that is both useful for the model and target distribution.

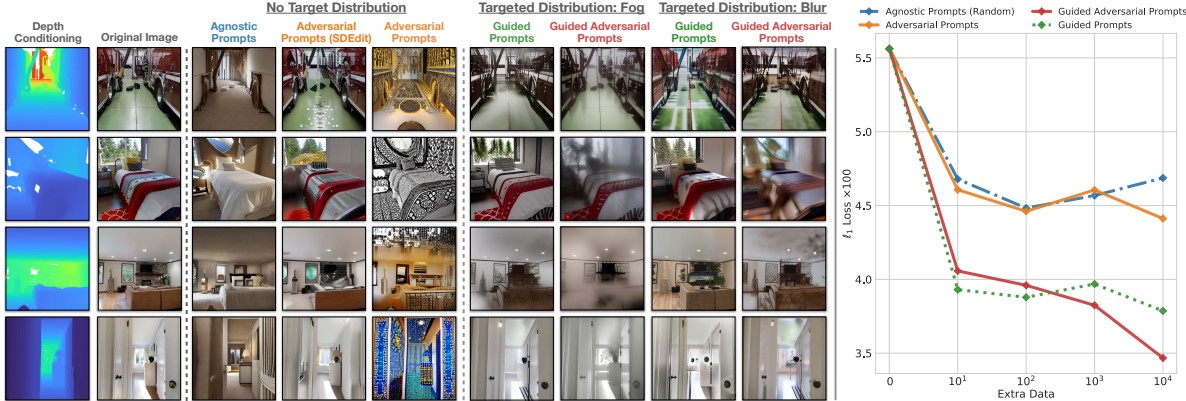

Figure 5: **Left: A comparison of generations with different prompts.** Generations with AP results in *diverse styles* that are *distinct from the original training data and with agnostic prompts.* AP (SDEdit) generations have slight modifications to the original image, as it was conditioned on them. The last 4 columns show the generations with text guidance for target shifts *fog* and *blur.* Compared to GP, GAP results in generations with more severe fog or blur. **Right: Performance of GAP with different amount of added data.** The target distribution here is *defocus blur* applied to Taskonomy test images. The plot shows the $\ell_1$ loss of the UNet model versus the amount of extra data generated and used for fine-tuning. The plot shows that both GP and GAP were able to guide the optimization toward generating training data relevant to the distribution shift. See Fig. 18 for results on other corruptions and Appendix C.2.3 for results on $10^5$ data.

**Training details.** We consider the following pre-trained models for $f$: 1) a U-Net (Ronneberger et al., 2015) model trained on the Taskonomy dataset (Zamir et al., 2018b; 2020) and 2) a dense prediction transformer (DPT) (Ranftl et al., 2021) model trained on Omnidata (Eftekhar et al., 2021b). The adversarial optimization was done with AdamW (Loshchilov & Hutter, 2019), learning rate of $5.0 \times 10^{-4}$, weight decay of $1.0 \times 10^{-3}$. We set the early stopping threshold (mentioned in Sec. 3.2) to 0.08 for the UNet model and 1.0 for the DPT model. They were trained with $\ell_1$ and Midas loss (Eftekhar et al., 2021b) respectively. We perform a total of 30 runs to get 30 different Adversarial Prompts. Similar to classification, for efficiency, we use 5 denoising steps during optimization and 15 for the final generations, with the DDIM (Song et al., 2020) scheduler, which we found to be a good trade-off between quality and efficiency. See the Appendix C.1 for more details.

**Comparing the generated images with different prompts.**

Fig. 5-left shows the generations from the baselines and our methods. The generations with AP are diverse and have complex styles. Incorporating SDEdit results in slight modifications to the original image, as it was used for conditioning. The last four columns show the results of using text guidance for the target distribution shift *fog* and *blur* (see Appendix C.4.2 for a comparison between generations from image and text guidance). The generations for GP results only in a mild level of fog or blur. In contrast, GAP results in more severe fog and blur corruptions. The generations for GP, using only the guidance feedback, result only in a mild level of fog or blur. Finally, incorporating both feedback mechanisms in GAP results in generating images with corruptions that

Table 1: **Quantitative results for depth estimation.** $\ell_1$ errors for the depth prediction task for U-Net and DPT models on Common Corruptions (CC), 3D Common Corruptions (3DCC), and cross-datasets (CDS) like Replica. Results for CC and 3DCC are averaged over all distortions and severity levels. AP outperform baselines for all models and distribution shifts. As GAP requires optimizing for each shift individually, and the benchmarks contain $\sim$ 30 shifts in total, we only perform GAP for a few shifts (see Fig. 5 and Fig. 18 for results). See Tab. 7 for results on other baselines.

| | UNet ($\times 100 \downarrow$) | | | | DPT ($\times 10 \downarrow$) | |
|---|---|---|---|---|---|---|
| | Taskonomy | | | Replica | Taskonomy | |
| Shift | Clean | CC | 3DCC | CDS | CC | 3DCC |
| Control (No extra data) | **2.35** | 4.93 | 4.79 | 5.38 | 3.76 | 3.42 |
| Agnostic Prompts | 2.47 | 5.03 | 4.17 | 5.30 | 4.06 | 3.58 |
| Agnostic Prompts (Random) | 2.38 | 4.96 | 4.11 | 5.14 | 3.88 | 3.51 |
| Adversarial Prompts | 2.49 | 4.36 | 4.02 | 5.12 | 3.40 | 3.28 |
| Adversarial Prompts (SDEdit) | 2.59 | **4.20** | **3.88** | **4.96** | **3.35** | **3.25** |

are both relevant to the corresponding target distribution and more severe, making them more useful for improving the model.

**Quantitative results.** Tab. 1 shows the results from finetuning our method and the baseline on the different generated datasets. Adversarial Prompts improves the performance of the model under different distribution shifts for both the UNet and DPT models. Fig. 5 *(Right)* shows the results with Guided Adversarial Prompts for the *defocus blur* corruption against the amount of extra generated data used for finetuning. Guided Prompts and Guided Adversarial Prompts result in a large performance improvement, compared to Adversarial Prompts or the baseline with only 10 extra data points. This suggests that the guidance loss *successfully steered the generations toward producing training data relevant to the distribution shift.* See Fig. 18 for results on other corruptions. As the dataset size increases, GP shows signs of saturation, whereas GAP further improves the model, showing the benefit of the adversarial feedback mechanism.

### 4.3 Additional Analysis

**Are adversarial prompts found for one model specialized for that model?** We study 1) whether models benefit *the most* from data generated using their own feedback and 2) whether they benefit at all from using data generated for another model (compared to Agnostic Prompts.) Tab. 2 shows that in most cases, a model performs best when trained with data generated using its own feedback, which shows the usefulness of the model-informed feedback mechanism. At the same time, we find that in most cases, prompts found for another model still outperform Agnostic Prompts, suggesting that adversarial prompts result in useful transfer between models.

**Does multiple iterations of adversarial optimization further improve performance?** We define an *iteration* as one round of adversarial optimization, i.e. optimizing Eq. (1) or Eq. (3), generation and fine-tuning. Given that all of the above results were obtained with a single iteration, we aim to see if there are benefits in performing multiple iterations. We perform a total of 8 iterations keeping the total number of generated images, prompts, etc., the same as for a single iteration for comparison. Fig. 6 shows that the first iteration of the multi-iterations setting results in the largest performance improvement and eventually converges to the performance of the single iteration approach. Thus, we chose to perform a single iteration for our main experiments. See the Appendix C.4.1 for additional implementation details and results.

Table 2: **Transferability of adversarial prompts.** We show the results of different combinations of the finetuning model and model used to generate additional data via model-informed feedback. We report top-1 accuracy for iWildCam and $l_1$ loss for depth.

| iWildCam, Acc. ($\uparrow$) | | | Depth, $l_2$ ($\downarrow$) | | |
|---|---|---|---|---|---|
| Data\Model | ResNet | ViT | Data\Model | UNet | DPT |
| ResNet | **83.97** | 72.61 | UNet | **4.73** | **3.39** |
| ViT | 83.87 | **77.21** | DPT | 4.88 | 3.46 |
| Agnostic Prompts | 72.94 | 73.07 | Agnostic Prompts | 5.27 | 3.66 |

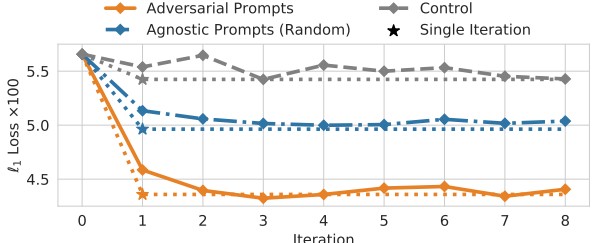

Figure 6: **Comparing the performance from running multiple iterations versus a single iteration** of adversarial optimization, generation, and fine-tuning. The plot shows the $\ell_1$ loss ($\times 100$) of the U-Net model against the number of iterations on the depth prediction task. The loss is computed on the Taskonomy dataset under common corruptions, averaged over all corruptions and severity levels.

## 5 Conclusion and Limitations

In this work, we aim to generate training data useful for training a supervised model by steering a text-to-image generative model. We introduced two feedback mechanisms to find prompts informed by both the given model and the desirable target image distribution. Evaluations on a diverse set of tasks and distribution shifts show the effectiveness of the proposed closed-loop approach in comparison to open-loop ones. Below we discuss some of the limitations of our work:

*Label shift:* In this work, we focus on generating novel images. However, some distribution shifts can also change the label distribution, e.g., for depth estimation, changing from indoor to outdoor scenes would result in a shift in depth maps. One possible approach could be learning a generative model over the label space (Le Moing et al., 2021) to control the generation in both the label and image space.

*Computational cost:* Estimating the gradient of the loss in Eq. (3) requires backpropagation through the denoising process of the diffusion model, which can be computationally demanding. Using approaches that reduce the number of denoising steps (Song et al., 2020; Luo et al., 2023) may be able to reduce this computational cost.

*Label Conditioning:* As discussed in Sec. 3.2, our method is limited by the faithfulness of the generation conditioned on the given label. For example, we found that the semantic segmentation ControlNet does not follow the conditioning accurately enough to be useful for the supervised model. Further developments in more robust conditioning mechanisms are needed to successfully apply our method to other tasks.

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

## A    Appendix Outline

We provide further discussions, details, and evaluations in the appendix, as outlined below.

- Appendices B and C describe additional implementation details for our classification and depth estimation experiments, respectively.

- Appendix B.6 describes an image guidance mechanism using Textual Inversion (Gal et al., 2022) and compares it with the CLIP guidance mechanism, on Waterbirds.

- Appendix C.2 provides additional quantitative results. See Appendix C.2.1 for comparisons with other data augmentation baselines for depth estimation and Appendix C.2.2 for comparisons with domain generalization baselines on iWildCam. See Appendix C.2.3 for results on training with more synthetic data.

- Appendices B.4.3, B.5.4 and C.3 provide **qualitative generations from all the Adversarial Prompts and Guided Adversarial Prompts** used in the Waterbirds, iWildCam, and depth estimation experiments.

  - Additionally, for depth estimation, we provide a qualitative comparison of Adversarial Prompts generations optimized on different models (UNet (Ronneberger et al., 2015), DPT (Ranftl et al., 2021)). For iWildCam, we also provide additional results using a ViT-B-16 (Dosovitskiy et al., 2020) instead of a ResNet50 (He et al., 2016b).

- Appendix C.4 provides additional analysis on the depth estimation experiments:

  - the single iteration vs. multi-iteration setting
  - a comparison of CLIP image and text guidance
  - **an assessment of the generalization of Adversarial Prompts from one model to another.**

## B    Implementation Details for Classification Tasks

### B.1    Training data generation

**Inpainting.**    As mentioned in main paper Sec. 3.1, for semantic classification tasks, we utilize the foreground object masks and use an in-painting technique proposed in Lugmayr et al. (2022) that preserves the masked region throughout the denoising process. In this section, we briefly describe this procedure and refer the reader to the original work for more details.

Let $m$ be a binary pixel mask, where a pixel is equal to 1 if the pixel contains the object and 0 otherwise, and $x$ be the original image from a training dataset. During generation, after obtaining a denoised sample $\tilde{x}_t$ at time $t$ we update it as $\tilde{x}_t \leftarrow m \odot x_t^{\text{orig}} + (1 - m) \odot \tilde{x}_t$, where $x_t^{\text{orig}}$ is the original image noised to have the correct properties of the expected Gaussian distribution at time $t$.

However, because we are using Stable Diffusion Rombach et al. (2022), the denoising process is done in latent space (using an encoder $\mathcal{E}$), not pixel space. This means that to apply inpainting, we must resize the mask $m$ to the latent space dimensions, and apply the above-described procedure in the latent space: $\tilde{z}_t \leftarrow m_z \odot z_t^{\text{orig}} + (1 - m_z) \odot \tilde{z}_t$, where $z_0^{\text{orig}} = \mathcal{E}(x^{\text{orig}})$ and $z_t^{\text{orig}}$ is its corresponding noised version. While this procedure usually performs well in preserving the original region of interest, we also paste the original masked region in the pixel space to obtain the final sample $\tilde{x} = m \odot x^{\text{orig}} + (1 - m) \odot \tilde{x}_0$.

**SDEdit Meng et al. (2021a).**    In addition to inpainting, depending on the setting, we also use SDEdit Meng et al. (2021a), a mechanism available to all diffusion models that allows to use an initial image to condition the generation of new images to be closer to the initial image. The mechanism is parametrized by the *SDEdit strength $s$*, which indicates the extent by which the model can deviate from the original image.

**Text-to-image model.** For our diffusion model, we use Stable Diffusion v1.5 [3].

## B.2 ALIA

Here, we give more details on the ALIA Dunlap et al. (2023) (Automated Language-guided Image Augmentation) baseline method, which aims at generating images targeting a particular test distribution similar to our guidance mechanism (main paper Sec. 3.3).

Given exemplar images from the test distribution, ALIA first captions each image using the BLIP Li et al. (2022) captioning model. Then, it uses the GPT-4 OpenAI (2023) LLM to summarize these captions into a list of domains asking it to produce descriptions that are agnostic to the class information. Dunlap et al. (2023) then use these prompts to generate additional training data. In order to preserve the original class information in their generations, they use SDEdit Meng et al. (2021a) or Instruct Pix2Pix Brooks et al. (2023). We refer the original paper for further implementation details. Below, we summarize resulting prompts we use for comparison in our results.

For Waterbirds Sagawa et al. (2019), we found that removing the prefix *"a photo of a {class name}"* from the original prompts when using the inpainting technique (Appendix B.1) to work slightly better for both the ALIA baseline and our CLIP text guidance (main paper Eq. (2)). We, therefore, use the following prompts:

- *"in a bamboo forest with a green background."*
- *"flying over the water with a city skyline in the background."*
- *"perched on a car window."*
- *"standing in the snow in a forest.",*
- *"standing on a tree stump in the woods."*
- *"swimming in a lake with mountains in the background.",*
- *"standing on the beach looking up."*

For iWildCam Beery et al. (2021), we keep the original prompts intact:

- *"a camera trap photo of a {class name} in a grassy field with trees and bushes."*
- *"a camera trap photo of a {class name} in a forest in the dark."*
- *"a camera trap photo of a {class name} near a large body of water in the middle of a field."*
- *"a camera trap photo of a {class name} walking on a dirt trail with twigs and branches."*

There are two main differences between ALIA and our method:

1. **The target distribution feedback**. ALIA aligns its prompts with the target distribution by utilizing captioning and summarizing. However, this summarizing process is not informed of the produced generations when using such prompts, and, thus, does not guarantee that the text prompt will accurately guide the generation process to images related to the target distribution.

2. **Model feedback.** ALIA is not model-informed. Thus, it doesn't necessarily generate images *useful* for training a given model.

Those two differences originate from the fact that ALIA is an **open-loop** method, i.e, it lacks the mechanism to refine the prompt based on the generated images. In contrast, our method uses model and target distribution feedback in a **closed-loop**. This allows our method to outperform ALIA and be more data-efficient.

---

[3]https://huggingface.co/runwayml/stable-diffusion-v1-5

Table 3: Generation and optimization parameters for Adversarial Promptsand Guided Adversarial Promptsfor classification tasks in Sec. 4.1.

| Dataset | SDEdit strength | denoising steps | guidance scale | # placeholder tokens | Opt. steps (AP/GAP) | $\lambda_t/\lambda_i$ |
|---|---|---|---|---|---|---|
| Waterbirds | 1. | 5 | 7.0 | 5 | 1000/1000 | 20/0 |
| iWildCam | 0.8 | 5 | 5.0 | 10 | 2000/10000 | 0/10 |

| Dataset | ALIA | | | Ours | | |
| | *SDEdit strength* | *sampling steps* | *text guidance* | *SDEdit strength* | *sampling steps* | *text guidance* |
|---|---|---|---|---|---|---|
| Waterbirds | 0.3 | 50 | 7.0 | 1. | 15 | 7.0 |
| iWildCam | 0.5 | 50 | 7.5 | 0.8 | 5 | 5.0 |

Table 4: Generation parameters for the classification experiments

### B.3 General implementation details

**Generation.** We report our generation parameters in Tab. 4. We use the DDIM Song et al. (2020) scheduler. We generate 384x384 resolution images. Those parameters were chosen based on visual inspection, ease of optimization and downstream performance (validation accuracy). **Training data.** After generation, ALIA's method consists of an additional filtering step to remove "bad" generations. This step relies on using a pretrained model to measure confidence on the generated images. However, given our method creates images that are adversarial to an iWildCam pretrained model, the filtering part of ALIA's pipeline is not usable on our data. Thus, to keep things comparable, we decided not to apply filtering both our method generated data and ALIA's generated data. However, it must be noted that Dunlap et al. (2023) only reports a 2% absolute accuracy drop between fitlering and no filtering on iWildCam (1.4% on Waterbirds), thus we do not expect a big difference in performance with ALIA's reported results and our results.

**Supervised Training.** We report our training parameters in Tab. 5. We use ALIA's codebase to finetune our models, which ensures fair comparison to the ALIA baselines. For everything except the generated data, the settings are the same as in ALIA. For both datasets, the starting model is a ResNet50 He et al. (2016a) model, pretrained on ImageNet Deng et al. (2009).

The reported test accuracy is chosen according to the best checkpoint, measured by validation accuracy.

### B.4 Waterbirds

#### B.4.1 Dataset details

Fig. 7 demonstrates the shift between train and test distributions in the Waterbirds dataset Sagawa et al. (2019). We follow the setting suggested in Dunlap et al. (2023) and use 1139 images as $\mathcal{D}_{\mathrm{tr}}$, where waterbirds appear only on water background and landbirds on land background. We add additional 839 examples either from the original dataset, where waterbirds appear only on land background and landbirds on water background ("Real OOD data"), or generated by Stable Diffusion with prompts obtained by one of the methods. For the data-efficiency plots (e.g., Fig. 3) we reduced the number of added examples by a factor of {1/2, 1/4, 1/8, 1/16}.

Since the original Waterbirds dataset does not provide masks for the exact generated images, we used the SAM Kirillov et al. (2023) segmentation model to obtain bird segmentation masks for training images. We use these masks to condition the generative model on the class by using inpainting as described in Appendix B.1.

| Dataset | Training | | |
| --- | --- | --- | --- |
| | *learning rate* | *weight decay* | *epochs* |
| Waterbirds | 0.001 | 1e-4 | 100 |
| iWildCam | 0.0001 | 1e-4 | 100/20 |

Table 5: Training parameters for the classification experiments

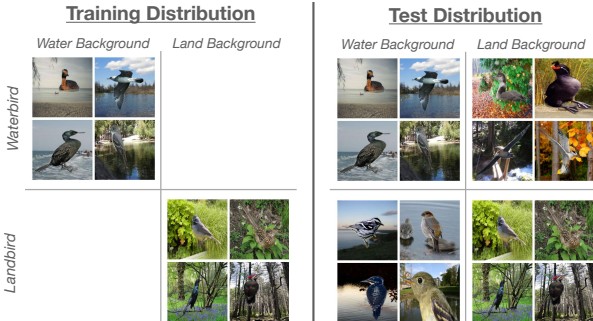

Figure 7: **Distribution shift in the Waterbirds dataset.** The background is a perfectly predictive spurious feature on the training distribution, but loses predictive power in the test distribution.

### B.4.2 Implementation Details

**Adversarial Optimization.** For adversarial feedback, we use the model trained only using the original training data $\mathcal{D}_{\text{train}}$ with complete spurious correlation. It is taken from ALIA checkpoints[4]. As the task is the binary classification, we use the cross-entropy loss for the opposite class as the adversarial loss: $\mathcal{L}_{\text{adv}}(\tilde{x}, y) = \mathcal{L}_{\text{x-ent}}(f(\tilde{x}), 1 - y)$, assuming $y \in \{0, 1\}$. This is equivalent to the negative cross-entropy loss referred in the text. We find four prompts per each class, i.e., eight prompts in total. Each prompt is composed of five new learnable tokens. We perform adversarial optimization for 1000 steps with learning rate 1e-3 using Adam Kingma & Ba (2017). We use five denoising steps during adversarial optimization and generate images for training with 15 steps. We do not use SDEdit for Waterbirds. See Tab. 4 for summary.

**CLIP Guidance.** For Waterbirds, we use CLIP text guidance by encoding each of ALIA's summarized prompts (see Appendix B.2) with the CLIP text encoder as described in main paper Sec. 3.3. In addition, we renormalize the averaged target text embedding to have the norm equal to the mean norm of the original prompts, and use the resulting vector as the target $e_{\text{t}}$. We use $l_2$ guidance loss: $\mathcal{L}_{\text{t}}(E_{\text{t}}(c_w), e_{\text{t}}) = \|E_{\text{t}}(c_w) - e_{\text{t}}\|_2^2$. We use $\lambda_{\text{t}} = 20$ and $\lambda_{\text{i}} = 0$ (i.e., no image guidance).

### B.4.3 Additional Qualitative Results.

In Fig. 8 and Fig. 9, we show a few generations using all 8 prompts used in the Waterbirds experiments for Guided Adversarial Prompts and Adversarial Prompts, respectively.

### B.5 iWildCam

### B.5.1 Dataset details.

The original iWildCam Beery et al. (2021) dataset is subsampled to create a 7-way classification task (background, cattle, elephant, impala, zebra, giraffe, dik-dik). The training set has 6,000 images with some classes having as few as 50 images per example. There are 2 test locations that are not in the training or validation set. Additionally, given $h$, the hour at which an image was taken, we define an image to be during

---

[4]https://api.wandb.ai/files/clipinvariance/ALIA-Waterbirds/y6zc932x/checkpoint/ckpt-Waterbirds-none-filtered-resnet50-1-0.001-0.0001/best.pth

Landbird                              Waterbird

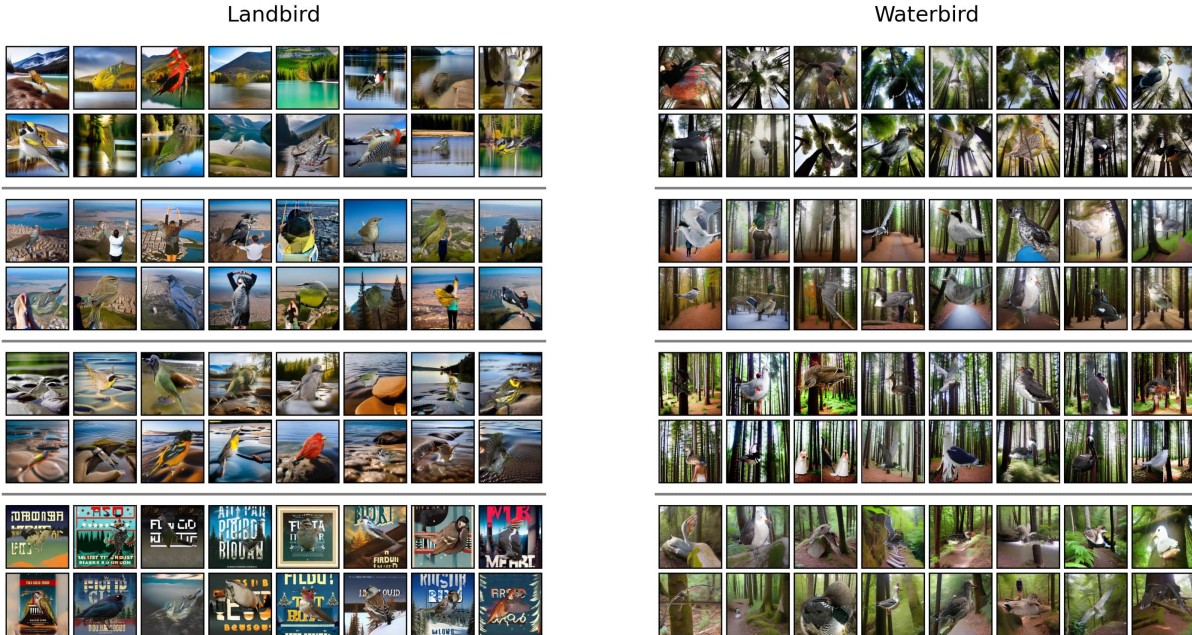

Figure 8: **Generation examples for Guided Adversarial Prompts**. Each column shows generations for the corresponding class. Each row in a column (separated by gray lines) shows generations for one found token. There are eight tokens in total, four for each class. GAP tend to generate landbirds on water (**left**) and of waterbirds on land (**right**), the combinations not present in the original training data (see Fig. 7)

"daytime" if $9 \leq h \leq 17$, and "nighttime" if $h \leq 5 \vee h \geq 20$. As said in main paper Sec. 4.1, for image CLIP guidance, we separate the target test locations into four groups (`location={1,2}`, `time={daytime, nighttime}`). We provide visualisation of the test locations (at day & night) in Fig. 10. For more details on the iWildCam subset construction, we refer to Dunlap et al. (2023) Section 8.3. For inpainting, the object masks are obtained from MegaDetector Beery et al. (2019).

### B.5.2 Alignment collapse solution for iWildCam.

As mentioned in main paper Sec. 3.1, choosing $\mathcal{L}_{\text{adv}}$ to be the negative cross entropy loss, i.e. minimizing the probability that the model predicts $y$, may not be the best choice. Indeed, given we use a random sample of 64 images to create our target embedding for the image CLIP guidance, the likelihood that animals were present on these 64 images is very high. This means that the target embedding, although mostly containing the "location concept", also partly contains an "animal concept". This means that the image CLIP guidance does not explicitly forbid the generation of new animals. Combined with optimizing the negative cross entropy loss, this leads to **adversarial animal insertions** at generation time, where a new animal of class $\hat{y}$ appears alongside the original animal of class $y$, destroying the $(\tilde{x}, y)$ alignment. In Fig. 11, we provide qualitative examples for this behaviour. To counter this behaviour, we choose $\mathcal{L}_{\text{adv}}$ to be the "entropy" loss, or uncertainty loss. More precisely, this loss is equal to the cross entropy loss where the target label $y$ is replaced by the soft label $\tilde{y} = [\frac{1}{|\mathcal{Y}|}, \cdots, \frac{1}{|\mathcal{Y}|}]$, the uniform distribution over all classes. This loss explicitly encourages generations that either (1) do not contain new animals (2) contain new animals that are not accounted for in the label space $\mathcal{Y}$.

Landbird                          Waterbird

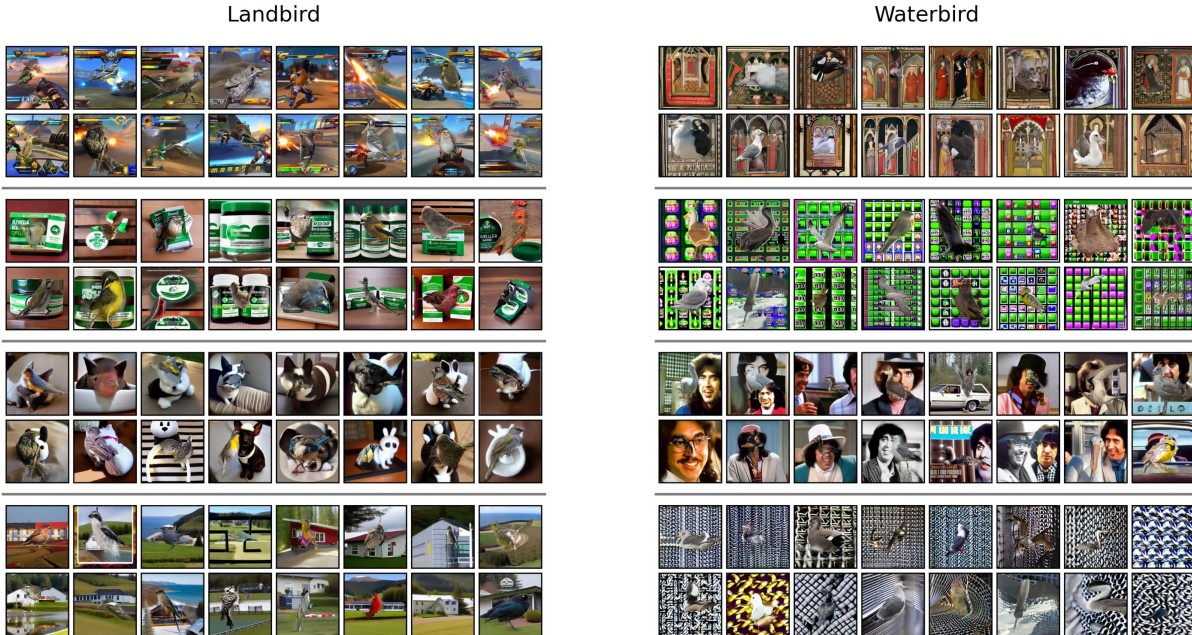

Figure 9: **Generation examples for** Adversarial Prompts. Each column shows generations for the corresponding class. Each row in a column (separated by gray lines) shows generations for one found token. There are eight tokens in total, four for each class. While AP finds tokens that fool the model, the generated images are different from the target distribution (land or water background).

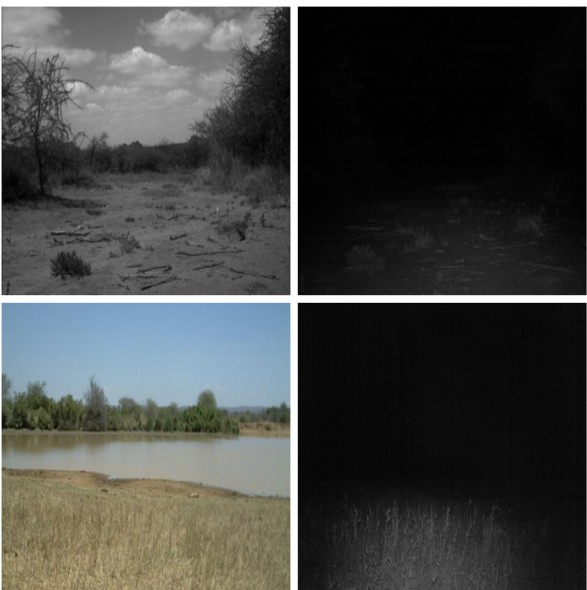

Figure 10: **iWildCam Test Locations.** Random samples from the four target distributions. First row is LOCATION=1, day & night. Second row is LOCATION=2, day & night.

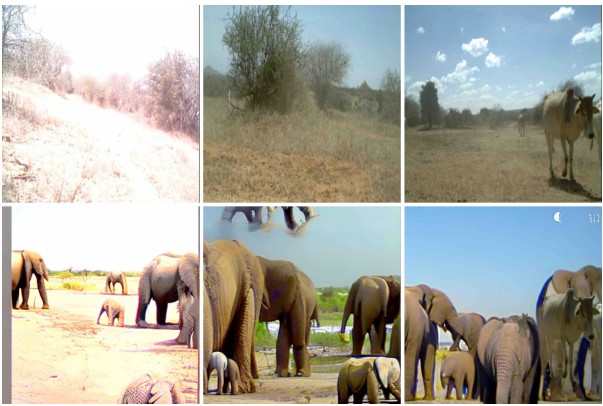

Figure 11: **Using the negative cross-entropy loss may lead to adversarial animal (e.g., elephants) insertions, destroying the alignment between $\tilde{x}$ and $y$.** First row contains the original training images. The labels are [*background*, *background*, *cattle*]. Second row contains the corresponding generated samples using a guided adversarial prompt, optimized with negative cross-entropy loss as the adversarial loss.

### B.5.3 Implementation details

**Adversarial optimization.** We describe here the parameters and settings used for optimization. If not precised, the same parameters were used for Adversarial Prompts and Guided Adversarial Prompts. As said in main paper Sec. 4.1, we optimize 4 prompts. Each prompt is composed of 10 placeholder tokens.

For optimization, we use a constant learning rate of 0.001, and a batch size of 8.

We use the "entropy" loss, described previously. For adversarial prompts, we train for 2000 steps. For guided adversarial prompts, we use CLIP guidance coefficient with $\lambda_i = 10$ and and $\lambda_t = 0$ (i.e., no text guidance). We train for a total of 10000 gradient steps. However, we don't optimize the adversarial loss for the first 2000 steps to allow the prompt to first converge to the target distribution region. One run takes about 3 hours, using one A100 GPU with 80GB of RAM, on our internal cluster.

For adversarial prompts, to generate 4 different prompts, we simply change the seed. For guided (adversarial) prompts, each prompt is w.r.t a new location & time of the day of the test distribution.

**Training data.** The generation settings are the same as the ones used during adversarial optimization. For each target domain-guided adversarial prompt, (i.e. location & time of the day), the source images (used to condition the generation with an object mask and through SDEdit Meng et al. (2021a)) are only images that match the time of the day of the target domain used during generation. Furthermore, for each prompt, we only generate one image per source image.

For ALIA, for each prompt, we generate one image per source image, from the whole training dataset. For the generation settings, given we use a slightly different generation process (inpainting) compared to their original implementation, we search ALIA's best-performing generation parameters (according to validation accuracy) over SDEdit strength [0.4, 0.5, 0.8] and guidance scale [5.0, 7.5]. We found the best-performing parameters for ALIA to be the same as the one reported by ALIA in their Github[5] i.e. SDEdit strength of 0.5 and guidance of 7.5.

**Finetuning.** The learning rate scheduler is a cosine scheduler, updated every epoch. The batch size is 128.

Our iWildCam pretrained model is taken from ALIA checkpoints[6]. ALIA trains the model from "scratch" (i.e. the model has never seen iWildCam data), for 100 epochs, on the combination of real + generated data. For our method, given we optimize the prompts based on a finetuned model feedback, it may not make as

---

[5]https://github.com/lisadunlap/ALIA
[6]https://api.wandb.ai/files/clipinvariance/ALIA-iWildCamMini/brr7b3ks/checkpoint/ckpt-iWildCamMini-randaug-filtered-resnet50-0-0.001-0.0001/best.pth

much sense to train the model from "scratch". Thus, we also introduce the variant where the iWildCam pretrained model is finetuned on the combination of real + generated data for 20 epochs, where finetuning means that every layer, except the last, is frozen.

For a fair comparison, both training settings are tested for ALIA and our method. We found that ALIA worked best when training from scratch and our method worked best when using the finetuning setting.

Finally, in their iWildCam experiment, ALIA fixed the number of extra generated points to be used in combination with real data during training to 2224 images. For the sake of comparison, we adopt the same limit in our experiments, with the added variant where the limit is 556 images, showcasing the data efficiency of our method.

### B.5.4 Additional Qualitative Results.

In Fig. 12 and Fig. 13, we show a few generations using each of the 4 Guided Adversarial Prompts and Adversarial Prompts used in the iWildCam experiments.

### B.5.5 Using ViT model.

In Fig. 14, we repeat the iWildCam experiment from the main paper (Fig. 4) with a ViT-B-16 Dosovitskiy et al. (2020) model. Additionally, we provide qualitative results for generations from adversarial prompts optimized with a ViT-B-16 model in Fig. 15.

### B.6 Image Guidance using Textual Inversion

In addition to the CLIP image guidance introduced in main paper Sec. 3.3, we also explore using Textual Inversion (TI) Gal et al. (2022) as an image guidance mechanism. Similar to the CLIP guidance, we use a few images $\{x_j\}$ from the target distribution. Now, instead of the similarity in the CLIP embedding space, we use the denoising loss between a generated image and one of the target images (see Eq. (2) in Gal et al. (2022)):

$$\mathcal{L}_{\text{TI}}(c_w) = \mathbb{E}_{x_j, z \sim \mathcal{E}(x_j), \epsilon \sim \mathcal{N}(0, I), t \sim U(0,1)} \tag{4}$$
$$\left[ \|\epsilon - \epsilon_\theta(z_t, t, c_w)\|_2^2 \right], \tag{5}$$

where $\mathcal{E}$ is the VAE Kingma & Welling (2022) image encoder and $\epsilon_\theta$ is the denoising UNet model from the Stable Diffusion model Rombach et al. (2022), and $x_j$ is sampled randomly from the set of available target images.

We test the TI image guidance on the Waterbirds dataset. We use the guidance loss from Eq. (4) with the weight 1000 (we found lower values to result in generations less faithful to the target images) and randomly sample 50 (unlabeled) images from the validation split of the original Waterbirds dataset where both types of birds appear on both types of backgrounds. We keep other settings the same as for GAP and AP.

Fig. 16 shows that TI guidance works on par with or better than CLIP guidance on the Waterbirds dataset. We found, however, that the TI guidance does not result in faithful generations for iWildsCam dataset, and further investigations are needed.

### B.7 Additional Analysis

We ablate hyperparameters like **1)** using $\ell_2$ or cosine loss for $\mathcal{L}_{\text{CLIP}}$, **2)** different ways of incorporating guidance e.g., text or image with CLIP or textual inversion (T.I).

The table shows the accuracy from different combinations of **1** and **2** on the Waterbirds dataset. Using $\ell_2$ and text guidance worked best, thus, we used this setting for our results in Fig. 3. T.I. compares generations to the original images in the pixel space, and Image in the CLIP embedding space, resulting in different guidance mechanisms and, hence, performance.

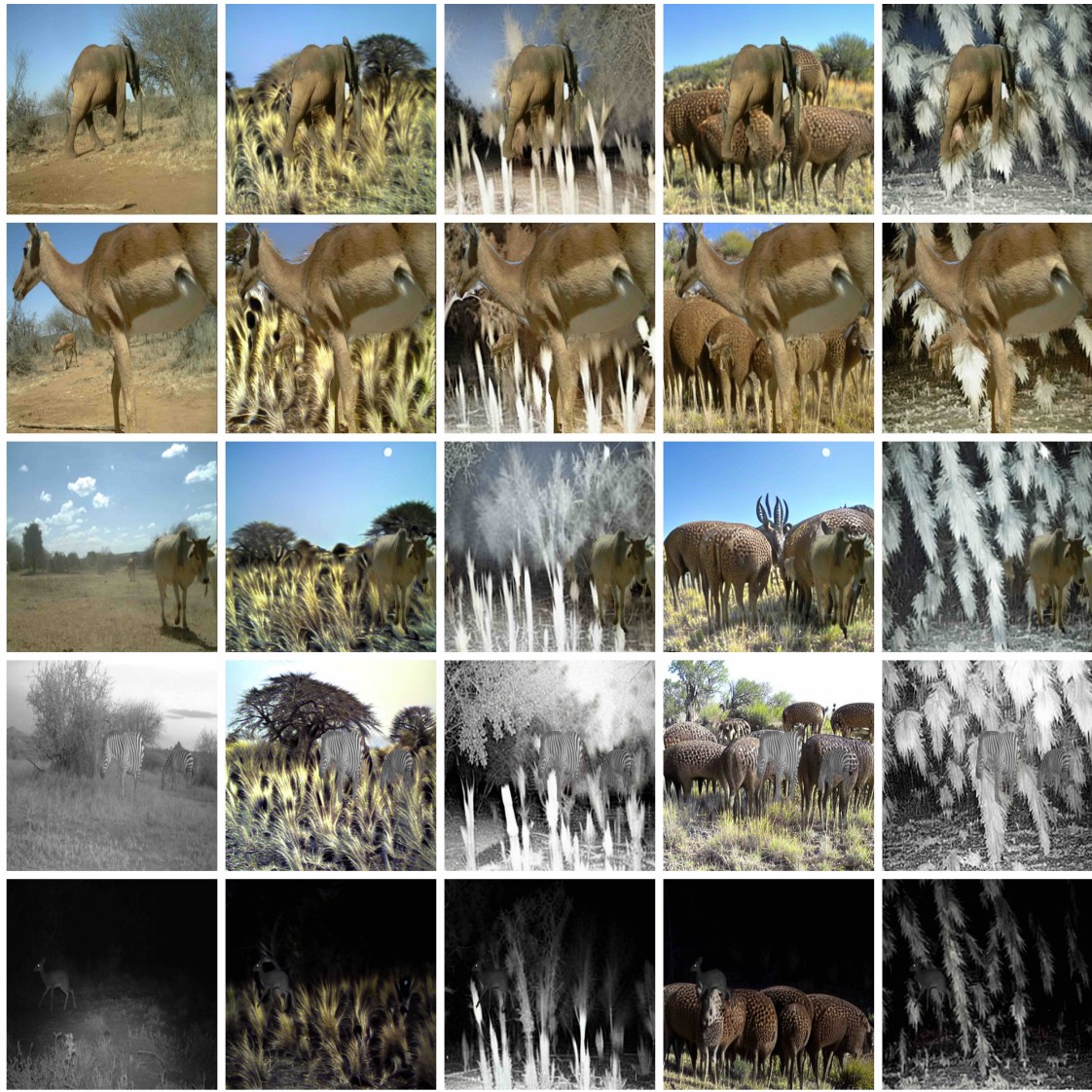

Figure 12: **Generations with the 4 guided adversarial prompts for iWildCam**. 1st column is the original image. Then from left from right, we have the guided adversarial prompts for LOCATION 1 during the day, during the night, and then LOCATION 2 during the day, during the night. SDEdit (strength 0.8) was used during the adversarial optimization and generation.

| Loss\Guide. | Text | Image | T.I. |
|---|---|---|---|
| $\ell_2$ | 84.0 | 77.3 | 83.6 |
| Cosine sim. | 82.5 | 72.7 | — |

Table 6: Results from different hyperparameters, namely, loss for CLIP guidance and how guidance is incorporated, on the Waterbird dataset.

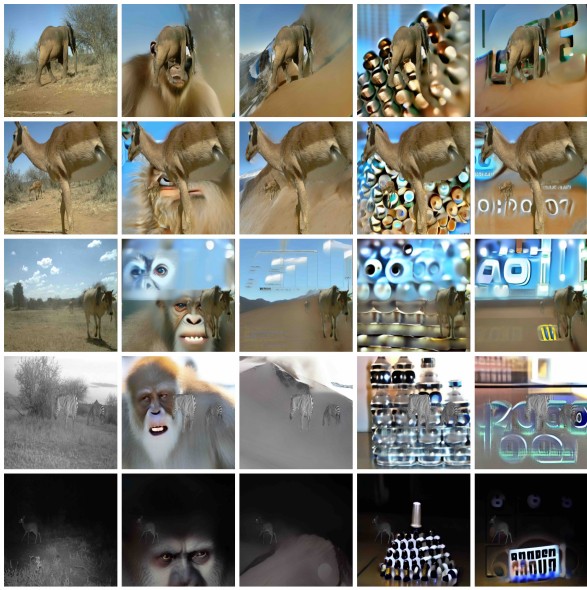

Figure 13: **Generations with the 4 adversarial prompts for iWildCam**. 1st column is the original image. Then, each column is an adversarial prompt initialized with a different seed. SDEdit (strength 0.8) was used during the adversarial optimization and generation.

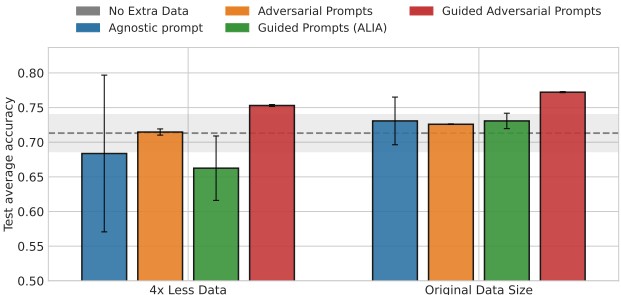

Figure 14: We train a ViT-B/16 model on the combination of the original training data and extra data generated using different types of prompts. We show the average accuracy on two iWildCam test camera trap locations. We run each experiment with three seeds and report the mean and standard deviation.

## C   Depth Estimation

### C.1   Depth training details

**Adversarial Optimization.** The adversarial optimization was done with AdamW Loshchilov & Hutter (2019), learning rate of $5.0 \times 10^{-4}$, weight decay of $1.0 \times 10^{-3}$, and batch size of 8. The token embeddings at the start of optimization are randomly sampled from $\mathcal{N}(\mu_{emb}, \sigma_{emb})$ where $\mu_{emb}$ and $\sigma_{emb}$ is the mean and standard deviation of all embeddings in the vocabulary. We set the early stopping threshold to 0.08 for the UNet model and 1.0 for the DPT model. **Note that these models were trained with different losses, $\ell_1$ for the former and Midas loss Eftekhar et al. (2021b) for the later.** Adversarial optimization is performed with the same loss as was used for training these models. One run takes about 15 mins using one 80GB A100, on our internal cluster. We perform a total of 32 runs, to get 32 Adversarial Prompts for the UNet model and 30 runs for the DPT model. As the DPT model was trained on Omnidata, which is a mix of 5 datasets, we have 6 runs for each dataset. Different number of placeholder tokens were also used for each run as suggested in Fig. 6 of the main paper. For the DPT model, we do 1, 8, 16 tokens runs for each dataset and also 3 runs with 32 tokens for each dataset. For the UNet model, 4 runs of 1, 8 and 16 tokens each and

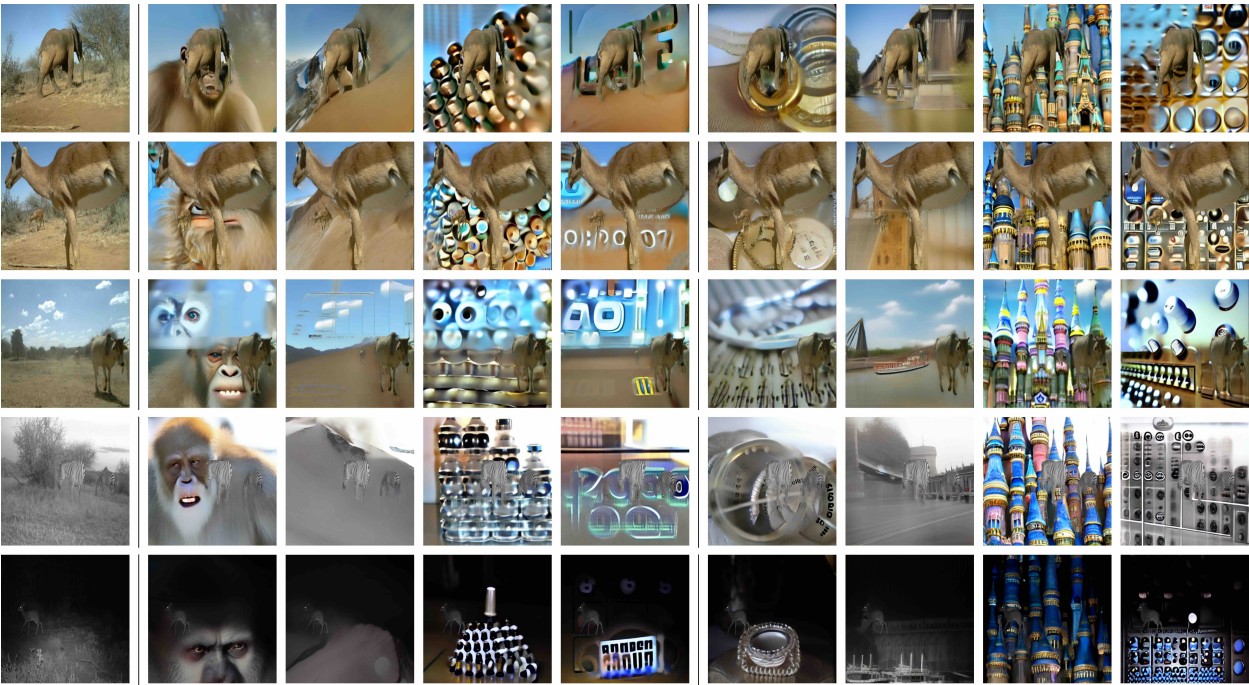

Figure 15: **A comparison of generations with adversarial optimization on different models.** First row is original image. Second to fith row are generated using the four ResNet-based adversarial prompts. The four last rows are generated using the four ViT-based adversarial prompts.

16 runs of 32 tokens were used, to get a total of 32 prompts. We also use a reduced number of denoising steps during optimization i.e., 5, as we found it to be more stable.

**Guided Adversarial Optimization.** The CLIP guidance coefficient for text and image guidance is set to 1 and 5 respectively. For image guidance, we randomly sampled 100 images from the target distribution. For text guidance, we used target distribution's name in the prompt, e.g., "fog" for the fog corruption from CC.

**Random prompts.** In our proposed method, we optimize for $n$ embedding vectors, resulting in a prompt, $c$. Thus, to match this setting, from a Gaussian distribution fitted on the embeddings from the vocabulary, we sample $n$ random embeddings to create a random prompt to be used in the data generation.

**Generation.** Generation is performed with the DDIM Song et al. (2020) scheduler and 15 sampling steps. We generate 80k images for the UNet model and 60k images for the DPT model for fine-tuning. For the GP runs with SDEdit, we used strength 0.6, for the GAP runs, strength 0.9. See

During optimization, we use only 5 denoising steps, as it is more stable.

**Fine-tuning.** For fine-tuning, we optimize the UNet model with AMSGrad Reddi et al. (2019) with a learning rate of $5.0 \times 10^{-4}$, weight decay of $2.0 \times 10^{-6}$ and batch size 128. For the DPT model, a learning rate of $1.0 \times 10^{-5}$, weight decay of $2.0 \times 10^{-6}$ and batch size 32.

## C.2 Additional Quantitative Results

### C.2.1 Depth Estimation

**Performance of non-SD baselines.** In Tab. 7, we show the results for depth estimation for two additional baselines, deep augmentation Hendrycks et al. (2021) and style augmentation Geirhos et al. (2018) that do not make use of generative models. Deep augmentation distorts a given image by passing it through an image-to-image model e.g., VAE Kingma & Welling (2022), while perturbing its representations. Style augmentation

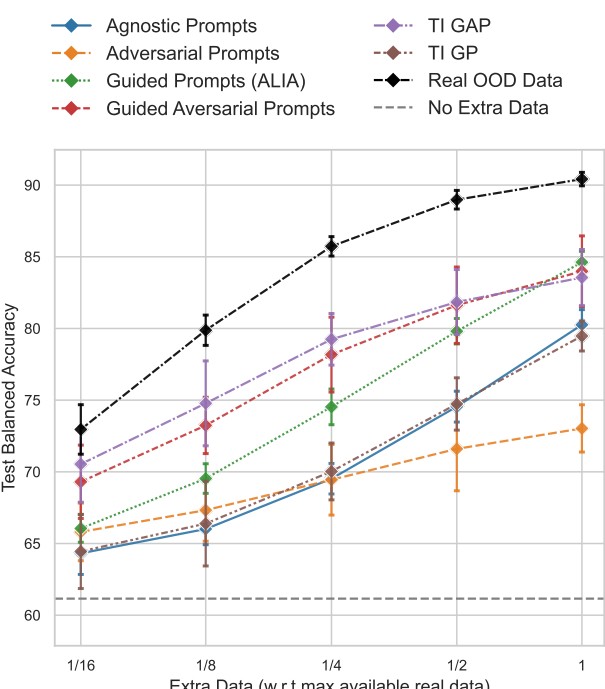

Figure 16: **Results of using Textual Inversion image guidance.** We use the same plot as in Fig. 3. We add two additional methods using Textual Inversion image guidance described in Appendix B.6. TI GAP is the Guided Adversarial Prompts with TI image guidance instead of the CLIP guidance. TI GP uses only TI guidance to find prompts. The TI guidance works on par with the CLIP guidance when used with adversarial optimization (TI GAP). However, using only TI guidance (TI GP) results in worse performance than using only text guidance with prompts found by the ALIA method Dunlap et al. (2023).

involves involves applying style transfer to the original training images. They perform comparably to Adversarial Prompts.

|  | U-Net | | | | DPT | |
|---|---|---|---|---|---|---|
|  | Taskonomy | | | Replica | Taskonomy | |
| Shift | Clean | CC | 3DCC | CDS | CC | 3DCC |
| Control (No extra data) | 2.35 | 4.93 | 4.79 | 5.38 | 3.76 | 3.42 |
| Agnostic Prompts | 2.47 | 5.03 | 4.17 | 5.30 | 4.06 | 3.58 |
| Agnostic Prompts (Random) | 2.38 | 4.96 | 4.11 | 5.14 | 3.88 | 3.51 |
| Adversarial Prompts | 2.49 | 4.36 | 4.02 | 5.12 | 3.40 | 3.28 |
| Adversarial Prompts (SDEdit) | 2.59 | 4.20 | 3.88 | 4.96 | 3.35 | 3.25 |
| Deep Augmentation | 2.42 | 4.24 | 3.70 | 5.01 | 2.83 | 3.70 |
| Style Augmentation | 2.42 | 4.15 | 3.85 | 5.16 | 2.80 | 3.10 |

Table 7: **Additional quantitative results on depth estimation.** $\ell_1$ errors on the depth prediction task for a pre-trained U-Net and DPT model. (Lower is better. UNet losses are multiplied by 100 and DPT losses by 10, for readability. Note that the two models were trained with different losses, thus their numbers are not comparable to each other.). We evaluate on distribution shifts from Common Corruptions Hendrycks & Dietterich (2019) (CC), 3D Common Corruptions Kar et al. (2022) (3DCC) and cross-datasets (CDS), Replica Straub et al. (2019). The results from CC and 3DCC are averaged over all distortions and severity levels on Taskonomy. Our method is able to generate training data that can improve results over the baselines on several distribution shifts. Generations with AP (SDEdit) gives better results than AP under distribution shifts. Thus, also conditioning on the original image seems to be helpful for these shifts. For the DPT model, the trends are similar, AP performs better than the baselines. Deep augmentation and style augmentation do not make use of a generative model for generating extra data. They perform comparably to AP.

### C.2.2 Classification with iWildCam

We show experimental results for several domain generalization baselines on iWildCam. The results for the synthetic data generation methods are based on the low-data regime, i.e., where we generate 4X less data. Our proposed methods, both AP and GAP, outperform these domain generalizaton baselines.

| Method | ResNet50 | GroupDRO (Sagawa et al., 2019) | ADA (Volpi et al., 2018) | SagNet (Nam et al., 2021) | L2D (Wang et al., 2021) | IRM (Arjovsky et al., 2019) | CausIRL (Chevalley et al., 2022) | Agnostic Prompts | AP | GP | GAP |
|---|---|---|---|---|---|---|---|---|---|---|---|
| Accuracy | 67.5 | 70.8 | 61.0 | 64.5 | 77.4 | 55.4 | 69.6 | 69.8 | 79.3 | 71.2 | 81.2 |

Table 8: **Additional quantitative results on iWildCam.** We compare our methods against several domain generalization baselines. GP and GAP still outperforms.

### C.2.3 Training with additional generated data

We experimented with training on more generated data for Waterbirds and Depth estimation. We found that the trends are dataset specific. For Waterbirds, GAP underperforms GP with 2X the amount of extra data (relative to the original training data, see Fig. 3 for the results for up to 1X extra data). However, for depth estimation GAP outperforms GP (see Fig. 5 for the results up to $10^4$ extra data).

| Dataset | Waterbirds (Acc.↑) | | Depth ($\ell_1$ Err.×100 ↓) | |
|---|---|---|---|---|
| Extra data | 1X | 2X | $10^4$ | $10^5$ |
| GP | 84.6 | 89.2 | 3.79 | 3.7 |
| GAP | 84 | 87.2 | 3.47 | 3.3 |

Table 9: **Training with additional data.** We extend the results in Fig. 3 and Fig. 5 to show the performance of our method with more generated data.

### C.3 Additional Qualitative Results

**Generations from all adversarial prompts & comparison of generations from different models.**
We show the generations from all Adversarial Prompts from the UNet model, without SDEDit (Fig. 19),
with SDEdit (Fig. 20), and multi-iteration (Fig. 21). Additionally, we provide the generations from two DPT
models, allowing us to assess the difference the model feedback has on generations. The first DPT model was
only trained on Omnidata (Fig. 22) and second was trained on Omnidata with augmentations from CC and
3DCC and with consistency constrains Zamir et al. (2020) (Fig. 23). The quantitative results in the paper
were reported only on the former DPT model.

There does not seem to be obvious differences in the styles generated between the two DPT models. However,
between the Adversarial Prompts from the UNet model with and without multi-iteration, the Adversarial
Prompts from the latter seems to result in much more diverse styles.

### C.4 Additional Analysis

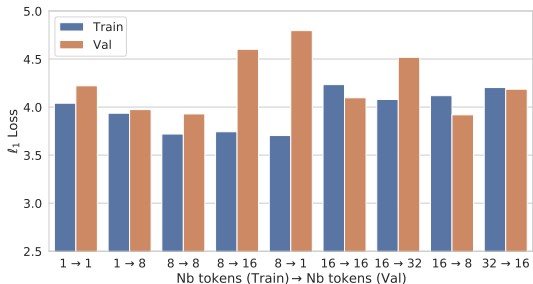

Figure 17: Generalization to generated images from similar or larger number of tokens. This plot shows the
performance of the UNet model when trained on Adversarial Prompts with $n$ number of tokens and tested on
Adversarial Prompts with $m$ number of tokens (denoted in the plot as $n \rightarrow m$). With the exception of $1 \rightarrow 1$
and $1 \rightarrow 8$, training on Adversarial Prompts with $n$ tokens and testing on $m, n \neq m$ results in higher loss
than training and testing on the same number of tokens.

#### C.4.1 Running multiple iterations of adversarial optimization vs a single iteration

Here, we provide additional analysis for the multi-iteration experiments in Fig. 6 in the main paper. We
optimize for 4 prompts in each iteration and noticed that if the number of placeholder tokens in a given
prompt is kept fixed throughout the iterations, the optimization to find new Adversarial Prompts becomes
more difficult. However, if we increase the number of tokens at each iteration e.g., 1 token per prompt for
1st, 8 per prompt for 2nd, etc, we are able to consistently find new Adversarial Prompts. Thus, we aim to
investigate the generalization of a given model to different Adversarial Prompts, e.g., is a model more likely
to generalize to Adversarial Prompts with the same number of tokens.

To perform this analysis, we generated data $D_n, D_m$ using AP with $n$ and $m$ tokens per prompt respectively
and measured the performance of a model fine-tuned on $D_n$ on $D_m$.

**Results for generalization to the same number of tokens.** In this setting, $n = m$, we use $n = m \in$
$\{1, 8, 16, 32\}$. For every $n$, we construct $D_n$ and $D_m$ to be generated using 4 Adversarial Prompts. We
fine-tune the model on $D_n$ and and validate both on $D_n$ and $D_m$ validation sets during the fine-tuning. The
results are shown in Fig. 17.

**Results for generalization to different number of tokens.** In this setting, $(n, m)$ are
$(1, 8), (8, 1), (8, 16), (16, 8), (16, 32), (32, 16)$ respectively. For every $n$ we fine-tune on $D_n$ and compute the
validation loss on $D_n$ and $D_m$. The results are shown in Fig. 17. As the loss for $n = m$ tends to be more

similar then when $n \neq m$, we chose to increase the number of tokens used per prompt in our multi-iteration setting.

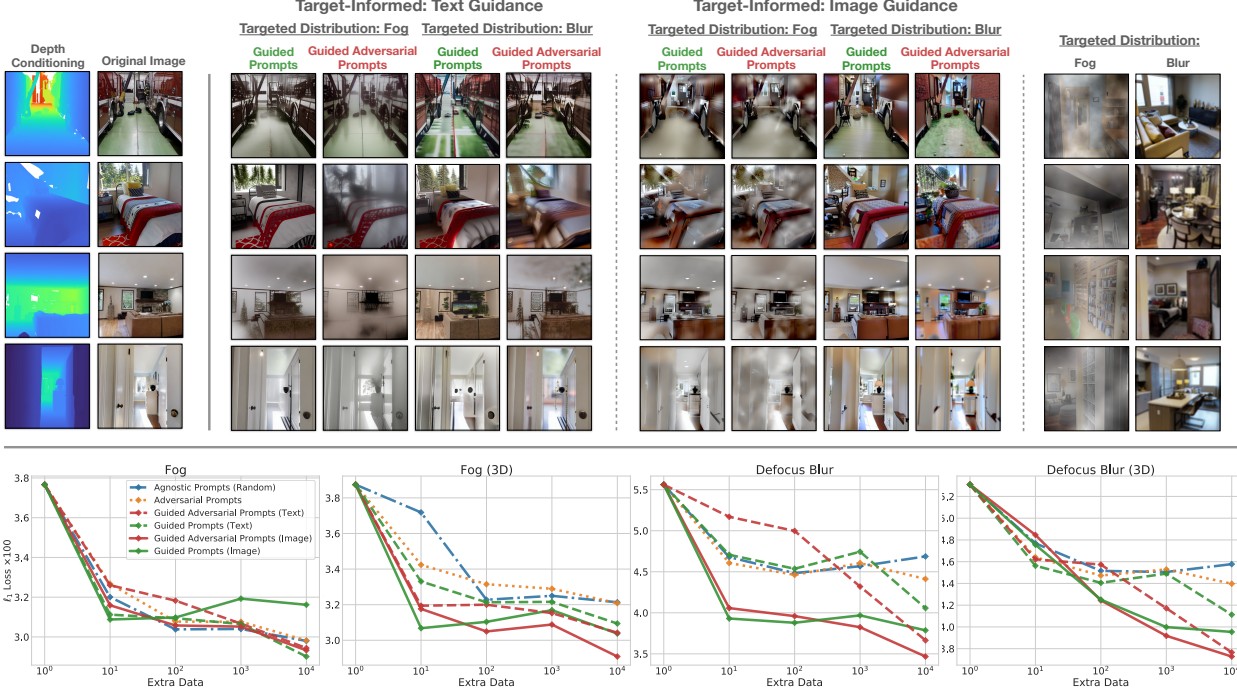

Figure 18: **A comparison of text and image guidance** for two distribution shifts fog and blur. The base model used here is the UNet model. **Top:** Generated images from Guided Prompts and Guided Adversarial Prompts. All images were generated with SDEdit, strength 0.9, thus, they tend to look similar to the original image (2nd column). Using Guided Prompts alone for either text or image guidance results in generations with a *mild* fog or blur. With Guided Adversarial Prompts, we get generations with *more severe* fog or blur. For image guidance, we sampled random (unlabelled) images with blur and fog from the Common Corruptions evaluation set. See the last two columns for some sample images. **Bottom:** The plots show the results from fine-tuning with text and image guidance, evaluated on fog and blur for CC and 3DCC. Note that the image guidance here uses (unlabelled) samples from the same target distribution that it was sampled on. In all cases, Guided Adversarial Prompts outperforms Guided Prompts with large enough extra data.

### C.4.2 CLIP image vs text guidance.

In Fig. 18, we compare the qualitative (top) and quantitative (bottom) differences in generations from text guidance and image guidance on defocus blur and fog. Note that image guidance uses sample (unlabelled) images from the corresponding target distribution that it is evaluated on, i.e., fog samples images from the fog corruption from the CC benchmark and fog (3D) samples images from the fog corruption of the 3DCC benchmark. If the target distribution name has (3D) appended to it, it is from the CC benchmark, otherwise it is from the 3DCC benchmark.

We observed some differences in the generations with text vs. image guidance (Fig. 18, top). Text Guided Prompts generates corruptions that are more realistic that image Guided Prompts. For example, fog gets denser further away from the camera or around the floor when text Guided Prompts are used for generations. For image Guided Prompts, as it was guided by the image samples from CC where the corruption is applied uniformly over the image, it learns to also apply a more uniform corruption over the image. We also noticed that the diffusion model was not able to generate certain shifts e.g., compression artifacts, we leave further analysis to future work.

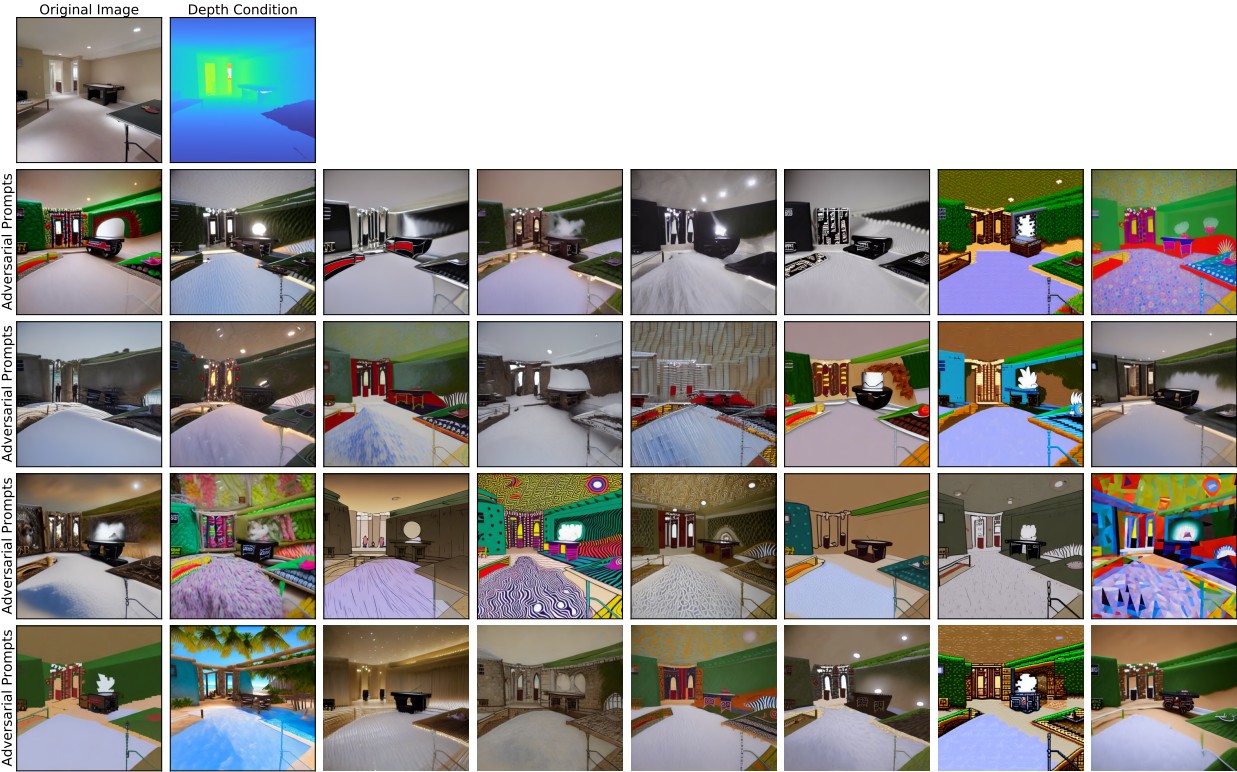

Figure 19: Generations from all Adversarial Prompts with the UNet model as the base model.

Quantitatively, we observed that image guidance tends to perform the best across the target distributions, with large enough extra data (Fig. 18, bottom).

### C.4.3 Generalization of Adversarial Prompts to different models.

We show how adversarial generations from Adversarial Prompts found for one model are for another model in Tab. 10. The generations from Adversarial Prompts found for e.g., the UNet model result in the highest loss when evaluated on the UNet model. However, the generations from Adversarial Prompts from the DPT model also result in similar loss. Similar trends hold for the DPT model. Thus, Adversarial Prompts found for one model are also able to result in high loss for another model.

| AP from\Eval on | Original data | UNet | DPT |
|---|---|---|---|
| UNet | 2.55 | 7.63 | 5.39 |
| DPT | 1.76 | 7.17 | 6.46 |

Table 10: Evaluation performance of a model on generated images from Adversarial Prompts from another model (without fine-tuning). For the UNet model we report the $\ell_1$ loss ($\times 100$ for readability) and the DPT model, the Midas loss ($\times 10$ for readability). The Adversarial Prompts attained from performing adversarial optimization on a UNet model and evaluated on the same model result in a loss of 7.63. Generations from the DPT model evaluated on the UNet model result in a loss of 7.17. Thus, the adversarial prompts found for one model seems to also be adversarial for another. We also report the loss on the original images for comparison.

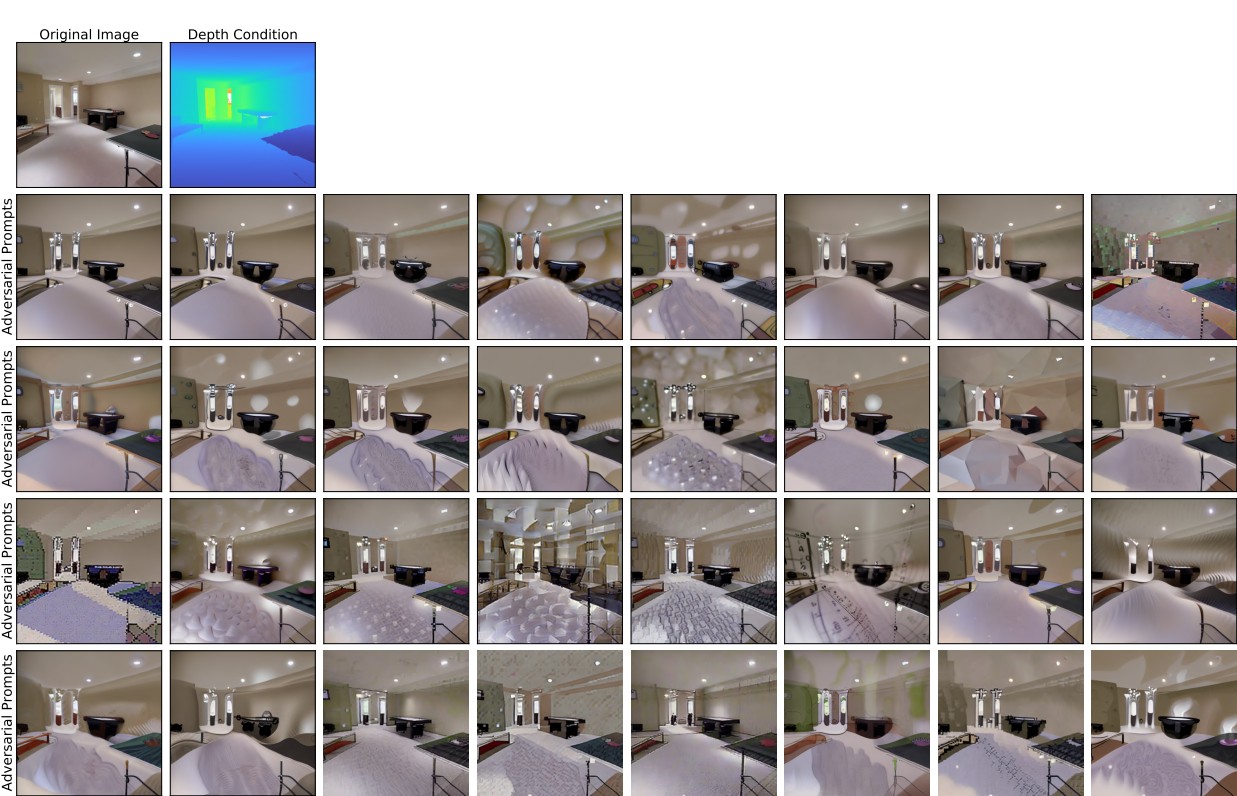

Figure 20: Generations from all Adversarial Prompts with the UNet model as the base model. SDEdit (strength 0.6) was used during the adversarial optimization and generation, thus, the generations look similar to the original image. Zoom in to see the different perturbations generated.

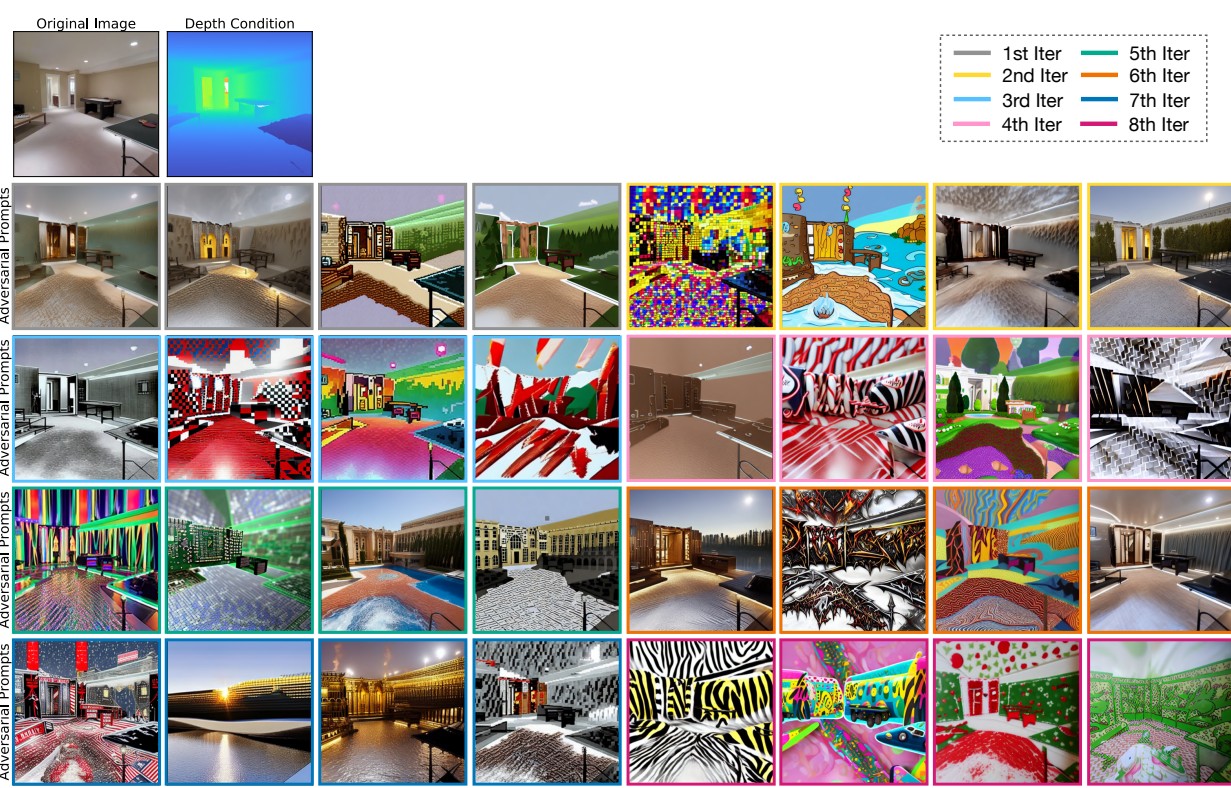

Figure 21: Generations from all Adversarial Prompts with the UNet model as the base model for the multi-iteration case i.e., multiple iterations of adversarial optimization, generation and fine-tuning. The colored borders denote the iteration number. Note that we set the early stopping threshold to be 0.1 for the first 3 iterations and 0.08 for the other iterations. We optimized for 4 prompts for each iteration, with an increasing number of tokens for each prompt.

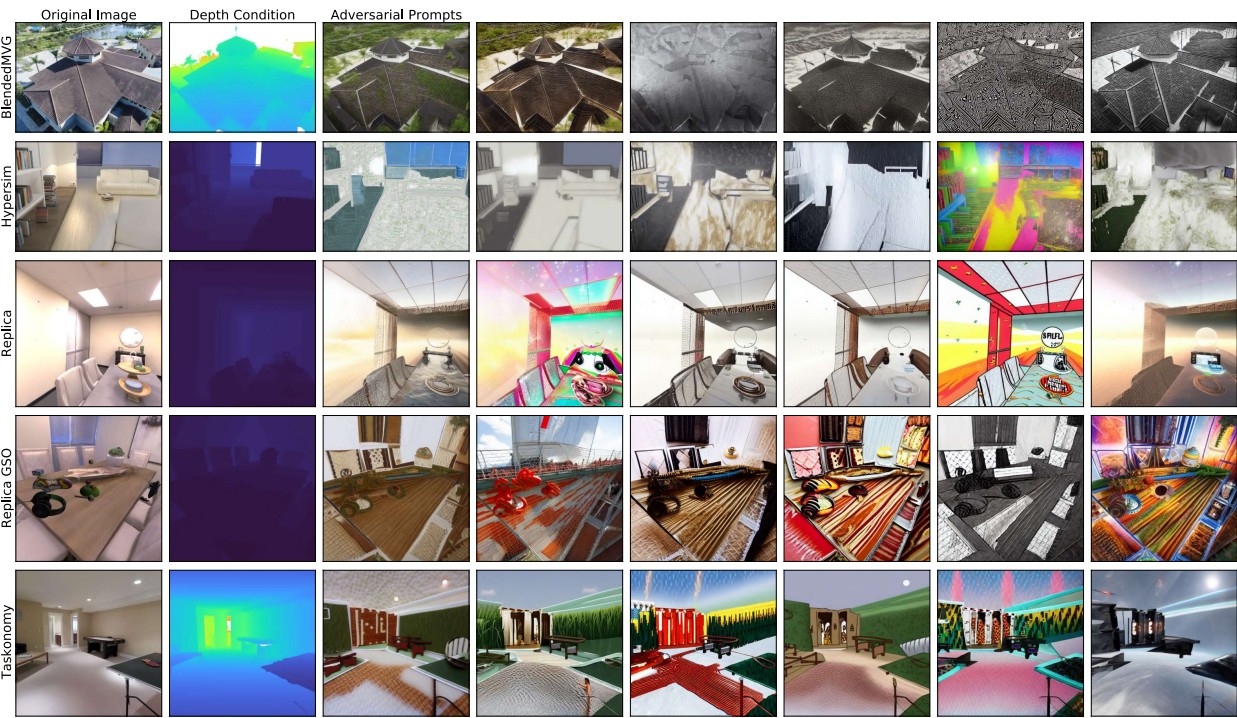

Figure 22: Generations from all Adversarial Prompts with the DPT model as the base model. The model was trained on Omnidata which consists of 5 datasets and we optimized for 6 Adversarial Prompts for each data. Each row shows the generation from the 6 different prompts for that dataset.

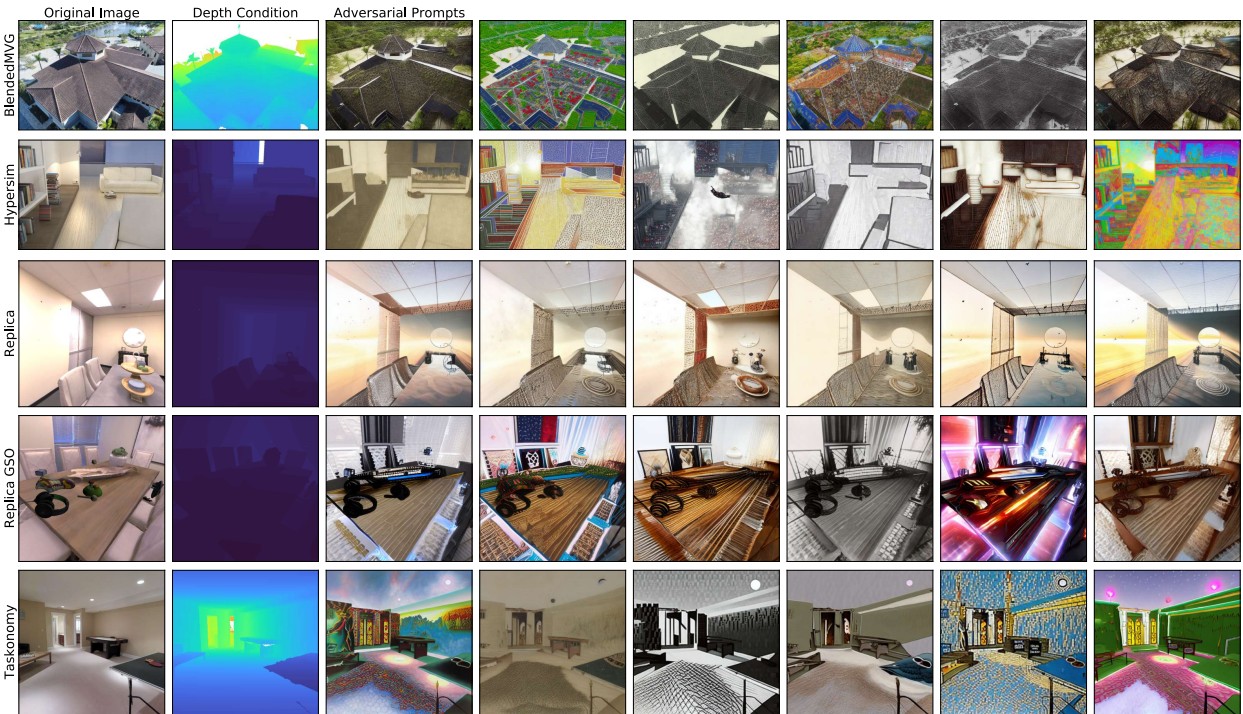

Figure 23: Generations from all Adversarial Prompts with a DPT model, also trained on Omnidata. However, this model was also trained with CC and 3DCC augmentations and consistency constraints.

## D  Broader Impact

We presented a method to control generative models to produce training data for supervised learning models. While our method is not particularly poised for negative use, it should be noted that powerful generative models are a general tool and the method has the potential to be used in ways authors did not intent. In addition, the data they are trained on may incorporate various societal biases or contain samples gathered in different ways from the internet. Furthermore, we will make our code and models publicly available. Thus, allowing for transparent inspection and safeguarding.

