# OpenReview forum: "Controlled Training Data Generation with Diffusion Models"
_TMLR — Accepted by TMLR_

### Review · Reviewer_HxDY · 2025-01-04

**Summary Of Contributions:**

This paper proposes a novel approach at generating additional synthetic training data to improve domain generalization of models. The technique learns prompts that encourage the generated images to (1) maximize uncertainty of the supervised model and/or (2) match the CLIP embedding of unlabeled target images or text descriptions of the target data.

**Audience:**

Yes

**Claims And Evidence:**

Yes

**Requested Changes:**

Necessary for Acceptance:
- Justification/experiments for why we would want to use GAP instead of generating more samples with a cheaper generation scheme. Also, it is important to show that this gap does not diminish when we continue increasing the amount of synthetic training data that we add.
- Additional baselines used in the domain generalization literature are important to include as baselines for the no extra training data setting

Nice to Have:
- Comparison to retrieving real images from the training datasets of models like Stable Diffusion
- Ablation using the mean of the top-k similar CLIP embeddings as the target for Eq. 2
- Qualitative analysis of the learnt prompts

**Strengths And Weaknesses:**

## Strengths
- The paper is well-written and clearly presented.
- The proposed approach is simple and intuitive

## Weaknesses
- From Figure 3, the performance difference between GAP and ALIA/Agnostic Prompting decreases considerably as more extra data is added. Does this trend hold if even larger amounts of additional data are used? Since ALIA/Agnostic Prompting avoids prompt learning and is thus less computationally intensive, one could potentially leverage more ALIA-generated data under a fixed computational budget. I believe that this is the biggest weakness of this work.
- While the baselines for different synthetic data generation methods are present, the methods used for training with no additional data could be improved. While the use of some types of "shallow augmentations" are explored, it would be great to also experiment with common domain generalization approaches such as GroupDRO which are specifically designed for these sorts of datasets
- Some recent papers [1] have demonstrated that training on real data retrieved from the training sets of models such as Stable Diffusion always outperforms training on data generated by these models. It would be interesting to see if the proposed method makes substantial progress towards closing this gap.
- In Eq. 2, the generative model is pushed towards generating examples that match the average CLIP embedding of the samples that we have for the target distribution. However, this will push all of the generated data towards a single target which can harm the diversity of the final generated distribution. It would be interesting to experiments by using the average clip-embedding to the top-k similar examples.
- While there is extensive qualitative analysis on the generated images, some qualitative analysis on the learnt prompts would be very interesting as done in [2] by mapping the continuous tokens to the nearest real word. It would be very interesting to see what the learnt prompts that generate the images shown in Figure 9 in the appendix

## References
- [1] The Unmet Promise of Synthetic Training Images: Using Retrieved Real Images Performs Better
 (https://arxiv.org/pdf/2406.05184)
- [2] Learning to Prompt for Vision-Language Models (https://arxiv.org/abs/2109.01134)

---

> ### Author Response · Authors · 2025-01-28
>
> > From Figure 3, the performance difference between GAP and ALIA/Agnostic Prompting decreases considerably as more extra data is added. Does this trend hold if even larger amounts of additional data are used?
>
> As requested, we experimented with generating more data for Waterbirds and Depth estimation. We found the trends are dataset specific. GAP under-performs with 2X data for Waterbirds but outperforms GP for depth.
>
> | Dataset | Waterbirds (Acc.↑) |  | Depth ($ℓ_1$ Err. ×100 ↓) |  |
> | --- | --- | --- | --- | --- |
> | Extra data | 1X | 2X | $10^4$ |$10^5$ |
> | GP | 84.6 | 89.2 | 3.79 | 3.7 |
> | GAP | 84 | 87.2 | 3.47 | 3.3 |
>
> > one could potentially leverage more ALIA-generated data under a fixed computational budget.
>
> ALIA uses a captioning model to attain domain descriptions which are then used in the prompts for generations. One key difference between ALIA and our method is that ALIA assumes that the captioning model’s description (1) accurately describes the image (2) that the description of the image captures what is important about the current task. Thus, ALIA may not work as well with tasks that involves more abstract concepts e.g., classifying tissue or cell samples. In contrast, our method is general as it does not rely on a captioning model and can be applied to tasks other than classification. We *learn the prompts* required for generation via an end-to-end process with feedback from the task network.
>
> Thus, under a fixed computational budget, if ALIA’s assumptions aligns well with the underlying task, we agree that using more ALIA-generated data may give better results. However, the strength of our method is that, due to the end-to-end optimization that learns the prompts, we ensure that we **generate synthetic samples that are relevant to the task**.
>
> > it would be great to also experiment with common domain generalization approaches such as GroupDRO which are specifically designed for these sorts of datasets
>
> GroupDRO uses group information e.g., for the Waterbirds dataset, whether the bird is on water or land, to minimize the loss of the worse-case group. In their training data, 95% of waterbirds are on water and 5% are on land (worst-case group). However, we following the experimental setup of ALIA where the bird is perfectly correlated with the background i.e., all waterbirds are on water. Thus, GroupDRO is not relevant for our experimental setting.
>
> We show experimental results for several domain generalization baselines on iWildCam. The results for the synthetic data generation methods are based on the low data regime i.e., where we generate 4X less data. Our proposed methods, both AP and GAP, outperforms these domain generalizaton baselines.
>
> | **Method** | ResNet50 | GroupDRO | ADA [1] | SagNet [2] | L2D [3] | IRM [4] | CausIRL [5] | Agnostic Prompts | AP |GP | GAP |
> | --- | --- | --- | --- | --- | --- | --- | --- | --- | --- | --- | --- |
> | Accuracy | 67.5 | 70.8 | 61.0 | 64.5 | 77.4 | 55.4 | 69.6 | 69.8 | 79.3 | 71.2 | 81.2 |
>
> [1] Volpi et al. Generalizing to unseen domains via adversarial data augmentation. NeurIPS’18.
> [2] Nam et al. Reducing domain gap by reducing style bias. CVPR’21.
> [3] Wang et al. Learning to diversify for single domain generalization. ICCV’21.
> [4] Arjovsky et al. Invariant risk minimization. arXiv 2019.
> [5] Chevalley at al. Invariant causal mechanisms through distribution matching. arXiv 2022.

---

> ### Author Response · Authors · 2025-01-28
>
> > Some recent papers [1] have demonstrated that training on real data retrieved from the training sets of models such as Stable Diffusion always outperforms training on data generated by these models.
>
> [1] focuses on classification task and retrieves from the LAION-2B dataset which consists of image caption pairs. In contrast, our method is general and can also be used for dense tasks such as depth estimation or semantic segmentation. **Retrieving from LAION-2B is not applicable for such tasks** as there we do not have an equivalent way of querying the dataset nor does it come with dense labels.
>
> For dense tasks, we use ControlNet for generation and ControlNet is trained with task specific data. However, these datasets are domain specific and orders of magnitude smaller than LAION-2B. For example, ControlNet with semantic segmentation was trained on COCO or ADE20K and is able to generalize to images more diverse than its training dataset. Thus, the training data of the diffusion model may not contain the relevant images that reflects the failure modes of the model or the target distribution shift.
>
> > In Eq. 2, the generative model is pushed towards generating examples that match the average CLIP embedding of the samples that we have for the target distribution. However, this will push all of the generated data towards a single target which can harm the diversity of the final generated distribution.
>
> The specific choice of the guidance mechanism using average CLIP embeddings could potentially lead to lower diversity. In general, if this is believed to be a problem, one can use a different guidance mechanism, e.g., one we explore in Appendix B.6. In addition, our considered main application scenario is to adapt to one specific test domain, which is more likely to be covered by nearby embeddings in the CLIP space. We also use a relatively small number of text descriptions (<10) or images (<100) describing the target domain.
>
> In Appendix B.6, we explore another guidance mechanism based on the Textual Inversion (TI) that does not have the described problem. It uses the original diffusion training loss and should, in theory, cover all modes of the distribution. In Fig. 16, we find that, **in practice, the CLIP guidance performs similarly to the TI guidance, which covers all the modes.**
>
> > qualitative analysis on the learnt prompts would be very interesting as done in [2] by mapping the continuous tokens to the nearest real word.
>
> We do not constrain the token embeddings to be close to that of the embeddings in the vocabulary during optimization (Eq. 1 and 2). We observed that the norms of the optimized token embeddings are much larger than the average norm of the embeddings in the vocab. Thus, projecting the continuous tokens into nearest real words do not result in meaningful prompts and generations look less adversarial i.e., more similar to images generated from agnostic prompts.

---

> > ### Comment · Reviewer_HxDY · 2025-01-30
> > **Response to authors**
> >
> > Thank you for the rebuttal.
> >
> > My primary concerns have all been adequately addressed. I would strongly encourage the authors to include a discussion on the benefits and drawbacks of ALIA and the domain generalization baselines somewhere in the paper.

---

### Review · Reviewer_busS · 2025-01-08

**Summary Of Contributions:**

The authors propose a method for generating image training data using diffusion models called guided adversarial prompts (GAP). Their main idea is to use adversarial generation to provide difficult training examples to learn better models, while at the same time use guides/constraints to ensure the generated adversarial examples fall within the target distributions. The empirical results are good and the proposed GAP method beats other state-of-art data augmentation methods using diffusion models. The major difficulty of the applying the proposed method is to ensure the generated adversarial examples are within the target distribution, which requires the use of different conditional generation mmethods and filtering, and can require different adaptations for different applications. Overall the paper is of good technical quality and I would recommend acceptance after minor changes.

**Audience:**

Yes

**Claims And Evidence:**

Yes

**Requested Changes:**

Address the weaknesses and questions above.

**Strengths And Weaknesses:**

- The main strength of the proposed method is the empirical improvements shown in a detailed set of experiments. The improvements over competing methods are significant as shown in Figures 3 and 4. There are also detailed analysis on the results in Section 4 and the Appendix.

- The main weakness in my opinion, also mentioned by the authors in Section 5, is the quality of conditional generation models. It is difficult to generate adversarial examples that match the label or target distribution constraints and one has to rely on tools like SDEdit or other heuristics. This makes the method more difficult to apply in practice and need to be adjusted in a case-by-case basis.

- Equation 1 is not clear. How are the weights w related to the continuous embedding c_{w_i}? Also, are there any constraints like norm constraints on the embeddings?

- One weakness of the proposed method is the possibility of (x, y) alignment collapse that the authors mentioned in Section 3.2. The current solution is to make it less adversarial by running fewer iterations but it does not seem satisfactory, nor does it guarantee that the generated image will follow the provided class label. The generated examples still need to be tuned by rules designed by humans.

---

> ### Author Response · Authors · 2025-01-28
>
> > Equation 1 is not clear. How are the weights w related to the continuous embedding c_{w_i}? Also, are there any constraints like norm constraints on the embeddings?
>
> The $c_{w_i}$ is a new token added to the vocabulary, while $w_i$ is its corresponding embedding. Concretely, instead of constructing a prompt as a sequence of discrete tokens $(t_1, \dots, t_n)$ and map them via CLIP’s (used in Stable Diffusion's conditioning) lookup embedding table to the continuous embedding space $(w_{t_1}, \dots, w_{t_n})$, which are then passed through CLIP’s text encoder, we introduce $n$ new embeddings $(w_1, \dots, w_n)$ that are directly passed to CLIP’s text encoder. This makes the generation process differentiable wrt $(w_1, \dots, w_n)$. There are no norm constraints on the embeddings.
>
> ### Alignment collapse
>
> > Difficult to generate adversarial examples that match the label or target distribution constraints.
> >
>
> > The current solution is to make it less adversarial by running fewer iterations but it does not seem satisfactory, nor does it guarantee that the generated image will follow the provided class label. The generated examples still need to be tuned by rules designed by humans.
>
> This is an issue for methods that trains on synthetic data. Most of these methods do post-hoc filtering (He et al., 2022, Dunlap et al., 2023), while we constrain the optimization. Like other methods, ours also introduces hyperparameters, e.g., an early stopping threshold. For example, for ALIA, they filter out generated images based on a confidence threshold in their post-hoc filtering step. This threshold also has to be tuned.
>
> Furthermore, for classification, our method makes use of the foreground object (which corresponds to the label $y$) masks and an in-painting technique (Lugmayr et al., 2022) that **preserves the masked region during generation**. This mask can be pseudo-labelled, and this technique ensures that the generated image will be aligned with its corresponding label y.
>
> Lastly, as also mentioned in Sec. 5, we believe that this is less of a problem for newer generative models.

---

### Review · Reviewer_uNsx · 2025-01-15

**Summary Of Contributions:**

This paper proposes a new mechanism called Guided Adversarial Prompts, which includes searching for adversarial prompts from supervised models as well as guiding generation to target distribution. The proposed method is in closed-loop manner and is designed for generating data  for specific distributions. The proposed method has been evaluated on both classification and dense prediction tasks, and results have shown it can help generate data for training purposes. The proposed method can serve as a powerful approach in the community and help mitigate data shortage in training generative models.

**Audience:**

Yes

**Broader Impact Concerns:**

In the paper, authors have not included any broader impact concerns. I would encourage authors to add one section, discussing relevant concerns with generated data and its process.

**Claims And Evidence:**

Yes

**Requested Changes:**

1. In the abstract, authors introduce the second mechanism as " Therefore, we introduce the second feedback mechanism that can optionally guide the generation process towards a desirable target distribution.". I recommend authors add 1 sentence introducing how the second feedback is processed, similar as the way introducing the first mechanism.
2. In Figure 1, it would be help to add what generative models used for illustrations?
3. In the subsection of "Generating aligned training examples." in Section 3.1, it is not clear to me how generating process works. Authors can clarify what do you use y or y&c to sample aligned examples. Do you discard texts when do sampling?
4. In Fig. 4, authors mentioned that "AP significantly outperforms GP in the low-data regime, suggesting that model-informed feedback generates more useful training examples.". I wonder do authors have qualitative analysis on it?
5. I recommend authors double check references across the paper, for example, in Table 1, it would be good to cite UNet and DPT here.

**Strengths And Weaknesses:**

1. The motivation of the paper is well explained for generating high-quality data within the target distribution so that the generated data can be utilized for training purposes.
2. Authors have explained the 2 mechanisms clearly in approach section.
3. Two tasks, classification and dense prediction for depths have been utilized to evaluate the proposed method. And experimental results are good.
4. Extensive details have been provided in the appendix which help understand the paper better.
5. Some weaknesses are listed in the below section.

---

> ### Author Response · Authors · 2025-01-28
>
> > In the subsection of "Generating aligned training examples." in Section 3.1, it is not clear to me how generating process works. Authors can clarify what do you use y or y&c to sample aligned examples. Do you discard texts when do sampling?
>
> We do not discard texts during sampling and both $y$ and $c$ are used to sample aligned examples. We generate new examples $\tilde{x} \sim g(c_w, y)$, where $y$ is the label to ensure $(\tilde{x}, y)$ is an aligned example that can be used for training and $c_w$ is the text prompt parametrized by $w$. $c_w$ is attained by optimizing the adversarial objective defined in Eq. 1.
>
> > In Fig. 4, authors mentioned that "AP significantly outperforms GP in the low-data regime, suggesting that model-informed feedback generates more useful training examples.". I wonder do authors have qualitative analysis on it?
>
> The qualitative results can be found in Fig. 4 and Fig. 13. Given that the optimization for AP does not constrain the token embeddings to be close to that of the original vocabulary, the generations may not be interpretable. The results in Fig. 4 shows that fine-tuning with small amounts of the data (4x less than the original dataset’s size) results in better performance compared to the baselines.
>
> #### Clarifications and references.
>
> Thanks for the suggestion, we will include them in the final version of the paper.

---

### Decision · Action_Editor_RYts · 2025-02-28

**Recommendation:** Accept with minor revision

**Comment:**

Authors are strongly advised to include discussions (regarding papers' exposition and explanations, asked by all reviewers) or additional materials (especially requested by reviewer HxDY) during the rebuttal phase.

**Audience:**

The problem definition is generally acceptable in the TMLR audience, and this paper introduces an interesting approach to handling guided image generation.

**Claims And Evidence:**

The authors propose a new approach for reliable dataset generation. It searches for adversarial prompts to properly guide the image generation process (called guided adversarial prompts (GAP)). The authors provide an extensive description of the proposed approach, and reviewers appreciated the results of the classification and dense prediction tasks.